

# The Met Office HadGEM3-ES Chemistry-Climate Model: Evaluation of stratospheric dynamics and its impact on ozone

Steven C. Hardiman[1], Neal Butchart[1], Fiona M. O'Connor[1], and Steven T. Rumbold[1,*]

[1]Met Office Hadley Centre, Met Office, FitzRoy Road, Exeter, Devon, EX1 3PB, UK
[*]now at: National Centre for Atmospheric Science, University of Reading, RG6 6BB, UK

*Correspondence to:* Steven Hardiman (steven.hardiman@metoffice.gov.uk)

**Abstract.** Free-running and nudged versions of a Met Office chemistry-climate model are evaluated and used to investigate the impact of dynamics versus transport and chemistry within the model on the simulated evolution of stratospheric ozone. Metrics of the dynamical processes relevant for simulating stratospheric ozone are calculated, and the free-running model is found to outperform the previous model version in 12 of the 14 metrics. In particular, large biases in stratospheric transport and tropical tropopause temperature, which existed in the previous model version, are substantially reduced, making the current model more suitable for the simulation of stratospheric ozone. The spatial structure of the ozone hole, the area of polar stratospheric clouds, and the increased ozone concentrations in the northern hemisphere winter stratosphere following sudden stratospheric warmings, were all found to be sensitive to the accuracy of the dynamics and were better simulated in the nudged model than the free-running model. However, significant biases in stratospheric transport, water vapour and ozone concentrations still exist in the nudged model. Further, stratospheric transport biases lead to biases in the downward ozone flux into the troposphere. Thus, whilst nudging can, in general, provide a useful tool for removing the influence of dynamical biases from the evolution of chemical fields, this study shows that nudged models still remain far from perfect.

## 1 Introduction

Previous studies have identified numerous couplings between ozone, greenhouse gases, tropospheric ozone precursors and stratospheric ozone depleting substances, and climate change. Increased carbon dioxide and near-surface ozone levels, for example, can impact vegetation and the strength of the land carbon sink (Sitch et al., 2007). Gas-phase constituents such as tropospheric and stratospheric ozone, have contributed to historical climate forcing (Stevenson et al., 2013; Myhre et al., 2013) and the inclusion of interactive chemistry, at least in some models, could affect estimates of climate sensitivity (Nowack et al., 2015). Likewise, climate change can impact on atmospheric composition through changes in the strength of the Brewer-Dobson circulation (Butchart and Scaife, 2001; Butchart et al., 2006), changes in methane lifetime (Johnson et al., 2001; Voulgarakis et al., 2013), changes in background and peak surface ozone concentrations (Fiore et al., 2012), temperature dependent chemical reaction rates (Waugh, 2009a), and the timescale for the stratospheric ozone layer to recover (WMO, 2011). Increasingly, there is also recognition of the extensive coupling between the troposphere and stratosphere, with stratospheric ozone recovery im-





pacting on tropospheric composition through stratosphere-troposphere exchange (e.g. Zeng et al., 2010) and photolysis rates (e.g. Zhang et al., 2014) and also impacting on surface climate (Morgenstern et al., 2009).

As a result, coupled chemistry-climate models have evolved to encompass both stratospheric and tropospheric chemistry coupled to state-of-the-art atmosphere-ocean climate models, in order that such couplings can be studied and fully understood.

Chemistry-climate models are also used to provide policy-relevant information, such as the assessment of strategies for mitigating and adapting to a changing climate with changing atmospheric composition (Eyring and Lamarque, 2012; Prinn, 2013). However, because of their inherent complexity, there is a strong need for comprehensive assessment and benchmarking of such models to sit alongside their development. In particular, the use of quantitative performance metrics (Waugh and Eyring, 2008) to both track the development of an individual model and/or to benchmark the performance of a multi-model ensemble

(Eyring et al., 2008), is important. These performance metrics have traditionally been used to consider how well individual model processes are simulated. In the present study, we take this further, considering the impacts of model processes on each other.

Nudging the dynamics of chemistry-climate model simulations towards observations is a technique used both to look at the impact of specific physical processes on atmospheric composition, and/or to remove the influence of unrealistic model

climatology from the evolution of chemical fields. Case studies covering just the length of a single observational campaign, and simulations covering long-term trends over the historical period, are both ways in which the use of nudged chemistry-climate models can enhance our understanding of the evolution of the chemical composition of the atmosphere. For example, Laat et al. (2001) consider the evolution of tropospheric ozone concentrations over the Indian Ocean during the spring of 1995, to evaluate the large-scale advection processes and associated tracer transport in their model. Dameris et al. (2005) consider the

impact of various "forcings" (including sea surface temperatures, volcanoes and the solar cycle) on chemical composition, to investigate which processes are well/poorly represented in models. Akiyoshi et al. (2016) present a case study of the evolution of chemical-species during the Stratospheric Sudden Warming of winter 2010, using both a nudged model and observations to study the structure in the chemical fields. A more general overview of the impact of nudging on chemistry-climate models is given in Jöckel et al. (2006, 2015), Telford et al. (2013), and Tilmes et al. (2016).

In the present study, the stratospheric dynamics, transport, and simulated ozone concentrations in free-running and nudged versions of the Met Office chemistry-climate model, HadGEM3-ES, are evaluated. The nudged simulations here make it possible to determine the ways in which biases in the model dynamical fields affect the accuracy of simulated stratospheric ozone concentrations, and thereby help attribute the remaining ozone biases to other components of the model (i.e. the transport and chemistry schemes).

This study is set out as follows. Section 2 describes the model setup and the simulations evaluated here. Section 3 presents the results, and is split into sections focusing on model metrics and the dynamics and ozone concentrations of the tropics and extratropics. Conclusions and discussion are given in Section 4.



## 2 Model setup and simulations

The Met Office model configuration used in this study is the chemistry-climate model HadGEM3-ES. The underlying atmosphere model is the Global Atmosphere 4.0 (GA4.0) configuration of HadGEM3 (Walters et al., 2014), and is based on the Met Office's Unified Model (MetUM). It has a horizontal resolution of 1.875° longitude × 1.25° latitude and 85 levels in the

5 vertical, covering an altitude range of 0–85km. This is coupled to the Global Land 4.0 (GL4.0) configuration of the JULES land surface model (Walters et al., 2014). For simulations requiring ocean and sea ice components, the Nucleus for European Modelling of the Ocean (NEMO vn3.4; Madec, 2008) ocean model, with a 1 degree resolution (ORCA-1) and 70 vertical levels, is used along with the Los Alamos sea ice model (CICE vn4.1; Hunke and Lipscomb, 2008).

This configuration represents a significant improvement in the physical model since the Met Office's contribution

(Morgenstern et al., 2010) to the Chemistry-Climate Model Validation activity 2 (CCMVal-2, Eyring et al., 2008). For example, the horizontal and vertical resolutions have increased from 3.75° longitude × 2.5° latitude and 60 vertical levels (model lid at 84 km). There have also been improvements to the atmosphere model physics and the addition of new ocean and sea ice components, all of which is documented in detail in Hewitt et al. (2011), Walters et al. (2011), and Walters et al. (2014). A significant result of these model improvements is the much reduced temperature bias at the tropical tropopause layer, which in

CCMVal-2 required the models based on MetUM to prescribe water vapour in this region. Water vapour is modelled interactively in the HadGEM3-ES simulations reported here.

This atmosphere-only or coupled atmosphere-ocean model HadGEM3 is, in turn, coupled to the gas-phase chemistry component of the United Kingdom Chemistry and Aerosol (UKCA) model (Morgenstern et al., 2009; O'Connor et al., 2014). The chemistry scheme is a combination of the stratospheric chemistry from Morgenstern et al. (2009) with the "TropIsop" tropo-

20 spheric chemistry scheme from O'Connor et al. (2014). Photolysis rates are calculated interactively using the Fast-JX scheme (Telford et al., 2013). Other aspects of the tropospheric chemistry configuration of UKCA that were not included in the Met Office's CCMVal-2 configuration, such as interactive lightning emissions (scaled to give 5TgN/yr), wet and dry deposition are now included as described in O'Connor et al. (2014). The interactive mass-based aerosol scheme (Bellouin et al., 2011) is unchanged from that used in CCMVal-2. Thus, the HadGEM3 model coupled to the UKCA chemistry scheme and the CLASSIC

aerosol scheme is referred to as HadGEM3-ES.

The results shown in this paper come from HadGEM3-ES simulations set up to follow the Chemistry-Climate Model Initiative (CCM-I) reference simulations (Morgenstern et al., 2016). These include an atmosphere-only historical simulation (REF-C1) and a coupled atmosphere-ocean historical and future simulation (REF-C2), which begin in 1960, as described in Eyring et al. (2013). The greenhouse gases (GHGs), ozone depleting substances (ODSs), tropospheric ozone precursor emis-

30 sions, aerosol and aerosol precursor emissions, sea surface temperatures (SSTs) and sea ice concentrations (for the atmosphere-only REF-C1 simulation), and the forcings from solar variability and stratospheric volcanic aerosol, are all as described in Eyring et al. (2013).

The coupled (REF-C2) simulation is spun up to 1960 conditions as follows. A 400 year spin up of the coupled atmosphere-ocean model to a perpetual pre-industrial state, is followed by a transient spin up of the coupled model, without interactive



chemistry, to 1950 conditions. Chemistry is then included, and a 10 year spin up to 1960 conditions is performed, as recommended by Eyring et al. (2013). For the atmosphere-only simulations, this 10 year spin up from 1950 with chemistry included (Eyring et al., 2013) is all that is required for the atmosphere to equilibrate.

Alongside the free-running atmosphere-only historical simulations (REF-C1), simulations in which temperature and hori-
zontal wind fields are nudged (Telford et al., 2008) towards the ERA-Interim reanalysis (Dee et al., 2011) are also run (REF-C1SD). McLandress et al. (2014) found that discontinuities in the upper stratospheric temperatures exist in ERA-Interim, in 1985 and 1998, due to changes in the satellite radiance data used. These discontinuities led to erroneous jumps in ozone concentrations in the upper stratosphere in their model, and therefore, in the "smoothed" nudged simulations detailed in Table 1, they were removed here using the technique of McLandress et al. (2014). To avoid introducing spurious noise, Merryfield et al.
(2013) found that the relaxation time scale must be longer than the time intervals between the reanalysis fields that are being nudged towards (6 hours for ERA-Interim) and noted in particular that relaxation time scales of 24 hours and 48 hours both gave good results (see their Figure 23). After some subjective trials, 24 hours and 48 hours were also found to be appropriate time scales for HadGEM3-ES, at least for the fields of interest here, and results using both time scales are included below. Nudging is applied over the vertical range 2.5km – 51km.
Details of these simulations are summarised in Table 1. Free-running simulations are run over the period 1960–2010 (REF-C1) and 1960–2100 (REF-C2), and nudged simulations are run over the period 1980–2010 (using initial conditions taken from REF-C1). As such, we analyse the period 1980–2010 in this study.

## 3    Results

### 3.1    Metrics

Metrics for evaluating the processes in chemistry-climate models relevant for the simulation of stratospheric ozone were developed as part of the CCMVal-2 project (Eyring et al., 2008). The metrics for dynamical processes are listed in Butchart et al. (2010, 2011). These dynamical metrics include one for the polar vortex final warming time but, for reasons explained later in this section, we choose to evaluate final warmings using the method of Hardiman et al. (2011), and thus this metric is not directly comparable and not included here. Table 2 lists the metrics used in this study.

Following the method of Waugh and Eyring (2008), "grades" are associated with each metric, to measure how accurately it is simulated, and these are calculated as follows:

$$g = 1 - \frac{1}{3}\frac{|\mu_{\mathrm{model}} - \mu_{\mathrm{obs}}|}{\sigma_{\mathrm{obs}}} \tag{1}$$

where $g$ is the grade assigned to the metric (and is set to 0 if calculated to have a negative value), $\mu_{\mathrm{model}}$ and $\mu_{\mathrm{obs}}$ are the model and observational mean values of the metric, and $\sigma_{\mathrm{obs}}$ is the interannual standard deviation of the observations (a
proxy for observational uncertainty). Thus, a value of 1 represents the model having an identical mean value to reanalysis (the "observations"), and a value of 0 represents the model mean value deviating by more than 3 standard deviations from





the reanalysis. Here we re-calculate these metrics for the Met Office model used in CCMVal-2 (UMUKCA-METO, REF-B1 simulation), using years 1980–2010 of the ERA-Interim reanalyses (Dee et al., 2011), instead of years 1980–2000 of the ERA40 reanalysis. These recalculated CCMVal-2 metrics can then be directly compared to those for all the free-running and nudged CCM-I simulations. Figure 1 displays these metrics in the same style as Butchart et al. (2010).

It is interesting to note that the UMUKCA-METO values for some of these metrics show a significant degradation compared to those given in Butchart et al. (2010) for the same simulation. Reasons for this are that:

   – the reanalysis dataset used here as the benchmark is ERA-Interim as opposed to ERA-40

   – analysis here is over the period 1980–2010 as opposed to 1980–2000 as used in CCMVal-2

In particular, using a different period can substantially alter the values of some metrics. For example, the PW_sh diagnostic
considers the variability in the heat flux and polar vortex temperatures in the southern hemisphere high-latitude winter. The sudden warming observed in 2002 (the only southern hemisphere sudden warming on record) significantly increases the overall variability in both these quantities. The semi-annual oscillation (measured by the SAO metric) increases in amplitude for the years 2000–2010, such that its mean amplitude for the period 1980–2000 is 15m s$^{-1}$ and this increases to 17m s$^{-1}$ for the period 1980–2010. This increase is not captured in the free-running simulations. The trend in mass upwelling in the tropical
lower stratosphere (measured by the up_70 diagnostic) is, for ERA-Interim, almost steady over the period 1980–1995, but shows a strong downward trend over the period 1995–2010, again not captured in the free-running simulations. This sensitivity shows a need to analyse over the full 30 years common to all simulations for calculation of the most reliable metric scores.

    Since reanalysis datasets and the period analysed will continue to be updated, there are issues with referring back to the values of metrics in previous reports (see also Austin et al., 2003). These issues could be minimized by

– using information from multiple reanalyses datasets as the metric "observations"

   – ensuring that the period analysed is of sufficient length to reduce the impact of interannual variability

where the "interannual variability" in this case is the interannual standard deviation of the observations, as noted above in equation 1. Of course, if possible, re-calculating metrics from older simulations and reports, using identical benchmark datasets and time periods for consistency, would allow for the cleanest comparison to the latest simulations. In any case, metrics
continue to provide an invaluable and concise indication of current model performance, indicating diagnostics where models are performing well and those where improvement is required.

    Comparing column 1 with columns 2 and 3 of Figure 1, the free-running version of HadGEM3-ES is shown to perform better than UMUKCA-METO in 12 of the 14 metrics (umx_sh and sao are the only exceptions). Further, as noted above, the SAO metric is particularly sensitive to the period analysed, so the differences in this metric between UMUKCA-METO and
30 the CCM-I simulations cannot be considered reliable. Thus, apart from the strength of the southern hemisphere polar night jet, the dynamics of HadGEM3-ES show improvements over the version of HadGEM used for CCMVal-2 (documented in Morgenstern et al., 2010).





As denoted in Figure 1 and Table 2, the metrics are divided into those that measure the mean climate of model simulations, and those that measure their variability. This division follows that in Butchart et al. (2010, 2011). Figure 1 demonstrates quite clearly that, whilst the nudged simulations (columns 4–7) are graded similarly to the free-running simulations (columns 2–3) in terms of mean climate metrics (an aspect in which the free-running model is already very good, though again with the exception of the southern hemisphere polar night jet strength), the nudged simulations outperform the free-running simulations in terms of variability.

The nudged simulations that use the discontinuity corrected ERA-Interim dataset (McLandress et al., 2014, columns 4 and 5 of Figure 1) show a better performance in the semi-annual oscillation metric than those without this correction (columns 6 and 7 of Figure 1), although given that the evaluation is against the unmodified ERA-Interim dataset it is unclear why this should be the case. Certainly it is expected that the only differences in performance between the nudged simulations with and without the discontinuities removed would be in the upper stratosphere (where the correction is applied) – a region assessed here only by the SAO metric.

The nudged simulations perform very well ($g > 0.9$) in almost all metrics, with the exceptions of tropical upwelling (up_70 and up_10) and the Quasi-Biennial Oscillation (qbo). Surprisingly, at both 70hPa and 10hPa the tropical upwelling in the free-running model is closer to the reanalysis than in the nudged model. Note, however, that due to the inherent noise and uncertainty in vertical velocities in reanalyses, vertical velocity is not nudged, only horizontal velocities. Furthermore, upwelling (or, more particularly, the residual circulation) may not be entirely due to dynamics, as previously thought, but perhaps also influenced by radiation (Ming et al., 2016a, b), something that is not constrained in any of the simulations (except indirectly, by nudging the temperature field). Indeed, some transport calculations (e.g. for descent in the polar stratosphere; Tegtmeier et al., 2008) use the diabatic rather than the kinematic vertical velocity (see Butchart, 2014). Thus, even though they use the same numerical advection schemes, the stratospheric transport in nudged simulations need not be more accurate than in free-running models, as discussed in more detail below. Note also that in both the free-running and nudged simulations the tropical upwelling at 10hPa is significantly closer to the reanalysis than is upwelling at 70hPa. This may be due to the model simulating a different structure of meridional circulation relative to that of the reanalysis (i.e. differences in shallow versus deep circulations; Birner and Boenisch, 2011).

The grading of the QBO metric below 0.8 for the nudged simulations is somewhat more surprising given that this metric depends only on zonal wind which *is* directly nudged. In fact, the nudged model accurately simulates the quasi-biennial oscillation in the zonal mean winds at 20hPa used in this metric, matching the reanalysis winds closely except not quite reaching the peak values of the oscillation and thus underestimating the amplitude of the relevant Fourier harmonics by 4% (not shown). However, since the power-spectrum approach inherent in this metric doesn't give a measure of uncertainty, this is calculated differently (by sub-sampling the data; Butchart et al., 2010). This produces an estimate of uncertainty that is small in magnitude and leads to this metric being very sensitive, and thus lower than might be expected in the nudged simulations. Caution is therefore needed when interpreting this metric for any model. Indeed, the sensitivity of this metric is only apparent due to the use of nudged simulations, thus demonstrating the importance of the nudged simulations for testing the robustness and reliability of metrics involving quantities that are directly nudged.



Figure 1 shows that, whilst there are small differences between the nudged simulations with 24 hour and 48 hour relaxation time scales, there are (with the exception of the SAO and heatflux metrics) no significant differences between the simulations using smoothed and unsmoothed datasets. From this point on, we will just consider the simulations using the smoothed dataset, with a particular focus on the 24 hour relaxation time scale integration ("REF-C1SD-24hr, smoothed").

Despite the issues caused by changing the reanalysis dataset and analysing over a different period, it is worth noting that, if a "direct" comparison is made, then values for the free-running CCM-I simulations (REF-C1 and REF-C2) are above the CCMVal-2 multi-model mean (Butchart et al., 2010) for 10 of the 14 metrics. The exceptions are the southern hemisphere jet maximum (umx_sh), tropical mean upwelling at 70hPa (up_70), and the tropical annual cycle (tann) and semi-annual oscillation (sao). Note also that, since the differences in the reanalysis dataset and period analysed cause the metric grades

of the Met Office CCMVal-2 model (UMUKCA-METO) to get worse (as already noted above), this adds confidence that the CCM-I model shows improvement over the CCMVal-2 model in terms of these metrics (assuming the differences when recalculating the grades of UMUKCA-METO can be considered representative of the CCMVal-2 multi-model mean).

## 3.2    Dynamics

Figure 2 shows climatologies of the annual mean zonal mean temperature and zonal wind in the REF-C1 simulation, and

biases in this simulation relative to ERA-Interim. A cold bias in the troposphere, and a warm bias at the tropical tropopause, which have existed in all the Met Office HadGEM models (Hardiman et al., 2015), exist also in the REF-C1 simulation, but these biases are small ($<$ 1K cold bias in the tropical troposphere, and a 1–2K warm bias at the tropical tropopause; Figure 2(b)). Also, as demonstrated in the metrics tmp_nh and tmp_sh in Figure 1, the biases in extratropical temperature at 50hPa are small ($\sim$ 0.5K in the northern hemisphere, and $\sim$ 1K in the southern hemisphere). Temperature biases of up to 8K do exist

in the upper stratosphere, but these are less important than biases at the tropical tropopause (which influence stratospheric water vapour) and the extratropical lower stratosphere (which affect Polar Stratospheric Cloud formation), and so will not significantly affect model performance. Figure 2(d) shows that the strong eastward jet bias seen at around 1hPa in the southern hemisphere (related to the poorly graded umx_sh in Figure 1) is accompanied by a westward bias just equatorward of the jet. This dipole structure to the bias is indicative of the jet being too strong because it is located too far poleward (possibly an issue

with the way in which non-orographic gravity waves are attenuated in the upper stratosphere; Scaife et al., 2002). These biases in temperature and zonal wind are, as expected, largely removed in the nudged simulations (Figure 1).

Figure 3 considers the seasonal cycle in temperature at 50hPa (relevant to polar stratospheric cloud formation during winter and spring) and zonal wind at 10hPa (a measure of polar vortex variability). Figure 3(a) shows that there are biases in the 50hPa temperature in both the northern and southern hemisphere high latitudes. The seasonal cycle in temperature is too weak in both

hemispheres, but this signal is more pronounced in the southern hemisphere, with up to a 4K warm bias seen in August. In both hemispheres, a warm bias of 1-2K is seen in polar spring. In the nudged version of the model, temperature biases are largely removed, with biases at 50hPa ranging from -0.88K to +0.10K (not shown).

Figure 3(b) shows that the winter polar vortex (at 10hPa) in both hemispheres is biased weak relative to the ERA-Interim reanalysis, consistent with the warm biases in the polar vortex shown in Figure 3(a). The weak bias is most significant in the



southern hemisphere winter, with a negative bias of up to 6m s$^{-1}$ in magnitude seen there. Again, this bias is removed in the nudged model, with biases in zonal mean wind at 10hPa showing magnitudes between -0.92m s$^{-1}$ and +0.66m s$^{-1}$. For both 50hPa temperature and 10hPa zonal winds, the biases in the REF-C2 simulation resemble those found in REF-C1, and hence are not shown. However, the magnitude of warm biases in the extratropical northern hemisphere is greater in REF-C2,

as discussed further below (see Figure 6).

### 3.2.1 Extratropics

A detailed look at the strength and variability of the zonal mean wind at 10hPa in both hemispheres (Figure 4) demonstrates that this is well simulated in the northern extratropics in all seasons, with the free-running models showing a small negative bias and slightly too much variability in October and November. However, the vortex strength and variability in southern

hemisphere winter and early spring are too weak in the free-running models. Despite this, the time of the vortex breakup, determined as the time when the zonal wind transitions from eastward to westward, is shown to be very accurately simulated in both hemispheres. Since the polar vortex acts as a barrier to transport, this vortex breakup allows transport of ozone into and out of the polar region, impacting springtime ozone concentrations in the high latitudes. Accurate simulation of the vortex breakup time is also important since the dynamical impact of the southern hemisphere extratropical stratosphere on the troposphere is

shown to be greatest during the time of the vortex breakup (Kidston et al., 2015).

Figure 5 shows this polar vortex breakup time at all altitudes for both hemispheres. This is accurately simulated in all simulations. The largest bias is seen in the northern hemisphere lower stratosphere for REF-C2 where the vortex breakup is around 10 days late, although even this is well within the 95% confidence interval for vortex breakup times calculated using ERA-Interim (Hardiman et al., 2011). As mentioned above, we do not include this metric in Figure 1 since we take a different

approach to that in Butchart et al. (2010), using instead an approach used in previous multi-model studies (Eyring et al., 2006). Hardiman et al. (2011) demonstrated that the time of the "final warming" of the polar vortex can be adequately calculated using monthly mean data in both hemispheres, and can be accurately calculated using monthly mean data in the southern hemisphere where the vertical profile of the final warming time is far simpler than in the northern hemisphere. In multi-model studies (the primary use of metrics) this has the advantages of requiring lower volumes of model data, and it also removes the noise

associated with daily data (something which is done in a less physically intuitive way, by using a low-pass filter, for the metric used in Butchart et al., 2010).

Of course, another important factor in determining the simulated heterogeneous ozone depletion, is the area of the Polar Stratospheric Clouds (PSCs). In this study, the size of the area in which temperature at 50hPa falls below 195K is used as a proxy for the PSC area (full details of how PSCs are simulated in HadGEM3-ES is given in section 2 of Morgenstern et al.,

2009). Figure 6(a) shows that the average October daily PSC area in the southern hemisphere extratropics is too low in the free-running model, consistent with the warm biases in the southern hemisphere extratropical temperatures at 50hPa shown in Figure 3(a). The average daily October PSC area across all years (1980-2010), in units of $10^6$km$^2$, is 0.9 in REF-C1, 1.6 in REF-C2, and 4.0 in both nudged simulations. The nudged simulations, as expected, show excellent agreement with ERA-Interim in this diagnostic. Thus PSC area in the free-running models is around 1/3 of the value as calculated from ERA-Interim



temperatures, and this is likely to have implications for heterogeneous ozone depletion. Figure 6(b) shows that, similarly in the northern high latitudes, the accumulated PSC area throughout northern hemisphere winter in the free-running models is, on average, around 1/2 the value it should be (according to ERA-Interim). There is substantial variability in the accumulated PSC area found in other REF-C1 and REF-C2 ensemble members (not shown or documented here) such that the large differences

in accumulated PSC area between the REF-C1 and REF-C2 simulations shown here lie within the expected variability. On average the CCMVal models were found to underestimate PSC area as compared to ERA40 (Butchart et al., 2011), and so this problem is not unique to HadGEM3-ES. Again, the nudged simulations show an accumulated PSC area that is in good agreement with ERA-Interim. Figure 6 (c) and (d) show minimum daily temperatures at 50hPa in the southern and northern high latitudes respectively, and show more clearly than the warm biases in the free-running simulations are somewhat larger

in the southern hemisphere winter than in the northern hemisphere winter, with warm biases of up to 4K seen in the southern hemisphere (consistent with Figure 3(a)). The variability in these minimum daily temperatures is shown to be too large in October and November in the southern hemisphere of the free-running simulations, but to be in good agreement with the reanalysis in the northern hemisphere in all simulations.

### 3.2.2   Tropics

Traditionally the Met Office climate model has suffered from a warm bias in the tropical tropopause region (Hardiman et al., 2015) leading to very high stratospheric water vapour concentrations. In HadGEM3-ES, however, this bias is relatively small (around 1–2K; see Figure 7(a)), leading to concentrations of water vapour (Figure 7(b)) that are only around 0.6ppmv too high in the stratosphere relative to MERRA (Rienecker et al., 2011)[1]. The remaining 1–2K bias in temperature is caused, in part, by simulated ozone concentrations that are too high (see Figure 17 below and also O'Connor et al., 2009; Hardiman et al.,

2015). The difference in 100hPa tropical temperature between REF-C1 and REF-C2 in January–May (Figure 7(a)) is localised to heights of around 150hPa–50hPa. Since this difference does not extend throughout the troposphere it is thought unlikely to be due to differences in sea surface temperatures per se (Hardiman et al., 2007). The same difference as that seen in 100hPa temperature is also seen in 70hPa water vapour concentrations (Figure 7(b)), though is delayed by 2 months consistent with the time taken for air parcels to rise from 100hPa to 70hPa in the tropics. Temperatures in the nudged model are inline with

observations (Figure 7(a)) leading to lower water vapour concentrations (Figure 7(b)). However, note that just nudging the temperatures and horizontal winds is not enough to remove any bias in water vapour concentrations (see also Hardiman et al., 2015) which are too low relative to the MERRA reanalysis (Figure 7(b)), and have an offset seasonal cycle, indicative of tropical upwelling that is too weak in the model (see Figures 9 and 10 below).

Accurate water vapour concentrations are very important for correctly simulating chemical species in the stratosphere,

including ozone. Water vapour, although not constrained in the nudged model, is strongly influenced by the cold-point temperature at the tropical tropopause. The annual cycle in cold-point temperature causes an equivalent annual cycle in water vapour concentrations entering the stratosphere in the tropics, and the upward transport of water vapour in the tropics gives rise to the so-called "tape-recorder" signal, shown in Figure 8. Due to an 8K warm bias in tropical tropopause temperature

---

[1]MERRA is used in Figure 7(b) as it is shown in Hardiman et al. (2015) to more accurately simulate water vapour concentrations than ERA-Interim.





in the UMUKCA-METO CCMVal-2 simulation (Morgenstern et al., 2010), stratospheric water vapour had to be prescribed in that model and the tape-recorder signal was therefore not simulated (Morgenstern et al., 2009). A significant improvement in the tropical tropopause temperature bias in HadGEM3-ES means that the tape-recorder is simulated in this model. The tape-recorder in the nudged (Figure 8(b)) and free-running models (Figure 8(c–d)) is compared against the Stratospheric Wa-

5 ter and Ozone Satellite Homogenized data set (SWOOSH – http://www.esrl.noaa.gov/csd/groups/csd8/swoosh/; Figure 8(a)). The tape-recorder signal appears more coherent far higher into the stratosphere in the nudged simulation. However, Figure 8(e) shows that this is not due to the amplitude of the annual cycle harmonic (the seasonal cycle in the tape-recorder signal) being greater in the nudged simulation than in the free-running simulations. A reduced amplitude in some of the sub-annual harmonics in the nudged simulation (not shown) may explain the increased coherence. Whilst water vapour concentrations

are slightly low in the mid-stratosphere of the nudged simulation (by 0.53ppmv at 30hPa), they are closer to observations in the lower stratosphere than in the free-running model. Water vapour concentrations are slightly high in the free-running model (by 0.44ppmv in REF-C1 and 0.57ppmv in REF-C2 at 30hPa). However, sensitivity experiments in a different version of the HadGEM3 model have shown changes in water vapour < 0.75ppmv to have no significant impact on the simulated stratospheric chemistry (not shown).

Whilst temperatures and horizontal winds are forced close to the ERA-Interim reanalysis in the nudged model, vertical winds are notoriously difficult to simulate accurately and are therefore not nudged. Figure 9 demonstrates that, as shown in Figure 1, nudging temperature and horizontal wind fields does *not* imply that the simulated vertical wind field will also be close to the reanalysis (and, further, there is reasonable agreement in the average magnitude of the vertical wind field across different reanalyses Butchart et al., 2011; Abalos et al., 2015). At some locations, the biases in residual vertical velocity in the nudged

simulations (Figure 9(b)) are of the same magnitude as the absolute values (Figure 9(a)).

Although the HadGEM3-ES simulations do capture the double-peaked nature of the 70hPa residual vertical velocity in the tropics (Figure 10(a)), like other models the peaks are too hemispherically symmetric (Butchart et al., 2010) and are biased low in both hemispheres. As a consequence, the upwelling mass flux from troposphere to stratosphere (Figure 10(b)), is too weak, particularly in the nudged simulations. Figures 10(a) and 10(b) show values of vertical velocity and upwelling, respectively, to

25 be around 20% lower in REF-C1SD-24hr than in the free-running simulations. This weak bias is much greater in the northern hemisphere winter (Figure 10(c)) than in the southern hemisphere winter (Figure 10(d)). Thus, Figures 9 and 10 show that the stratospheric circulation is very difficult to simulate accurately, even in nudged simulations.

An alternative diagnostic of the strength of stratospheric transport is the so-called "age of air" (Figure 11). The mean age of stratospheric air (Waugh, 2009b) denotes the time since that parcel of air was last in contact with the troposphere, and thus

gives an indication of the rate of transport to different regions within the stratosphere. Figure 11(a) shows that age of air is too old in the lower stratosphere in the tropics (by up to 0.5 years compared to age inferred from CO2 observations) – consistent with too little upwelling shown in Figure 10(b). However, age of air is too young throughout much of the stratosphere (Figure 11(b)), which cannot be explained by biases in upwelling from the troposphere to the stratosphere alone (Birner and Boenisch, 2011). Nonetheless, the age simulated by HadGEM3-ES represents a significant improvement on that seen in the Met Office

UMUKCA-METO CCMVal-2 simulation (Morgenstern et al., 2010), in which stratospheric air was 1–2 years too old. More-



over, the age simulated by HadGEM3-ES is in much better agreement with observations (Figure 11). Furthermore, Linz et al. (2016) argue that it is the latitudinal gradient in age of air, and not age itself, that best diagnoses the strength of the meridional mass circulation and that this gradient, at any height, is independent of the circulation above. This latitudinal gradient is much improved in the HadGEM3-ES model as compared to UMUKCA-METO. For example, at 21km the latitudinal gra-

dient ($(35° − 45°N)$ - $(10°S − 10°N)$) in HadGEM3-ES is 1.7 years, in line with the observations, whereas it is 3.2 years in UMUKCA-METO.

### 3.3 Ozone

Figure 12 shows time series of column ozone as simulated in the free-running and nudged models, compared to the Total Ozone Mapping Spectrometer (TOMS) satellite data (McPeters et al., 1998). Near-global (60°S–60°N) annual mean ozone (Figure

12(a)) is biased high relative to observations by around 10 Dobson Units (DU). Near-global ozone loss is slightly stronger in the nudged model than in the free-running model, such that near-global ozone concentrations in the nudged model agree well with the TOMS data after around 1990.

Figure 13(a) shows the global net annual mean stratosphere-troposphere-exchange (STE) of ozone (i.e. the net mass flux of ozone across the tropopause – see caption of Figure 13 for details). Consistent with Figure 10(b), which showed the tropical

mass upwelling from the troposphere to the stratosphere to be biased weak, the STE ozone flux in the model simulations is found to be too low as compared to ERA-Interim. Currently the best estimate of STE ozone flux inferred from observations is $550 \pm 140$ TgO3/yr (Olsen et al., 2001), thus even the ERA-Interim estimate of STE ozone flux is around 250 TgO3/yr too low. Figure 13 (b) and (c) show that, consistent with Figure 10 (c) and (d), the bias in STE ozone flux (as compared to ERA-Interim) is more prominent in the northern hemisphere winter than in the southern hemisphere winter. The similarity

between Figures 10 and 13 demonstrates the influence of the stratospheric meridional circulation on the STE ozone flux. A bias in STE ozone flux will have implications for extratropical tropospheric climate (see section 7.3 of Butchart, 2014), surface ozone concentrations (e.g. Zhang et al., 2014), and the global tropospheric ozone budget (Wild, 2007; Young et al., 2013).

### 3.3.1 Extratropics

The amount of ozone depletion in the extratropics is similar in all simulations (Figure 12(c,d)), and agrees well with the TOMS

observations. However, column ozone concentrations that are too high are indicative of an ozone hole that is too small in area. Further, we have seen 50hPa temperatures biased high in the free-running model (Figure 3(a)), PSC areas biased too low (Figure 6), and negative biases in the southern hemisphere polar vortex strength (Figure 4(b)). Figure 14 shows column ozone over the south pole in October, averaged over the years 1997–2002, as compared against the 220 Dobson Unit (DU) contour from the TOMS satellite data averaged over the same 6 years. Southern hemisphere extratropical column ozone is biased high,

by around 40DU, in all versions of the model (Figure 12(d)), leading to a simulated ozone hole (area with column ozone values below 220DU) that is too small. Hence an accurate simulation of PSC areas (Figure 6(a)) is insufficient to eliminate errors in the areal extent of the ozone hole, at least when the nudging is to ERA-Interim temperatures. On the other hand the nudging *does* remove errors in the orientation of the ozone hole which is slightly displaced from the pole (Figure 14). The phase of the



"croissant" shape in maximum ozone around 60°S is also more accurately simulated in the nudged model, with a minimum value around 50°W, in line with TOMS. In the free-running simulations, the location of the minimum varies from around 80°W to around 110°W.

Northern extratropical zonal mean column ozone concentrations are very well simulated (Figure 12(c)). In terms of azonal ozone structure, conclusions for the northern hemisphere (Figure 15) are the same as for the southern hemisphere. The amplitudes of the two ozone maxima simulated around 120°E and 140°W are similar in the free-running model (especially in REF-C2). In the nudged simulation, however, the amplitude of the 150°W maximum is far greater than that of the 120°E maximum, in closer agreement with TOMS. Biases in the zonal asymmetry of ozone (i.e. the "croissant" shape in the southern hemisphere, and larger maximum around 150°W in the northern hemisphere) arise due to corresponding biases in the amplitude and phase of the planetary stationary waves in the stratosphere which, again, are eliminated by the nudging. The fact that free-running models in general are unable to reproduce the correct phase (and amplitude) for the stationary waves (see Figures 8 and 9 of Butchart et al., 2011) makes it rather difficult to determine what phase to include when prescribing zonally asymmetric ozone forcings in models without interactive chemistry. The results here show that this will almost always lead to different ozone concentrations from those obtained by the same model using self determined ozone.

A further way in which dynamics influence ozone concentrations is through the enhanced poleward transport that follows Stratospheric Sudden Warmings (SSWs; Akiyoshi et al., 2016). Figure 16 shows the average positive ozone anomaly following a SSW, which increases ozone concentrations by around 15% compared to their climatological values. In the middle stratosphere where ozone is dynamically controlled the anomalies in the nudged simulation agree well with ERA-Interim but at higher levels where chemistry starts to dominate the anomalies are too large (c.f. Figure 16 (b,e) and Figure 16 (a,d)). Without nudging, the model on average underestimates the strength of the adiabatic temperature increase associated with the SSWs (c.f. Figures 16 (i) and 16(g)) and consequently the anomalously high polar ozone in the month following SSWs is weaker than observed (c.f. Figure 16 (c,f) and Figure 16 (a,d)). As well as SSWs influencing ozone, it is also the case that zonally asymmetric ozone can increase the frequency of simulated SSWs (Albers et al., 2013), thus creating the possibility for a feedback in models with interactive chemistry.

### 3.3.2 Tropics

The simulated interannual variability in tropical column ozone (Figure 12(b)), in both free-running and nudged simulations, agrees well with the observations. However, column ozone concentrations are again biased high, with average biases of 12.6DU in the free-running model and 7.0DU in the nudged model (Figure 12(b)). The largest biases, relative to TOMS, occur in December-January-February (Figure 17(a)). Whilst tropical temperature and water vapour concentrations can influence ozone concentrations, they are clearly not the only influences on simulated tropical ozone. Cold-point temperature is constrained to reanalyses in the nudged model and water vapour concentrations in the nudged model are too low relative to MERRA (Figure 7), yet ozone concentrations, although improved, are still too high even in the nudged model (Figure 17(a)). Figure 17(b) shows that this high bias primarily occurs in the tropical tropopause region (as shown also for the Met Office CCMVal-2 model





by Figure 7 of Gettelman et al., 2010), where convection, lightning emissions and biomass burning emissions also have an important influence on ozone concentrations (Stevenson et al., 2006), and thus the bias exists throughout the troposphere.

## 4 Conclusions

This study analyses the historical period (1980–2010) of free-running and nudged simulations using HadGEM3-ES, the Met Office chemistry-climate model as configured for inclusion in the Chemistry-Climate Model Initiative. In the nudged model configuration, the relaxation time scale of the applied nudging was found to be important (Merryfield et al., 2013) although it was not the case that a single time scale could be found in which all metrics were improved. In the present study, 24 hour and 48 hour nudging time scales were both found to give good results overall, for the stratospheric fields considered.

Metrics of dynamical processes relevant for the simulation of stratospheric ozone were calculated for all model configurations. These were compared against the metrics as re-calculated over the period 1980–2010 for the previous model configuration, UMUKCA-METO, used in CCMVal-2 (Morgenstern et al., 2010). The free-running model configuration is shown to have significantly improved since the UMUKCA-METO configuration, performing better in 12 of the 14 metrics considered here. The grades associated with some metrics were found to be sensitive to the reanalysis period used, implying that the period used should be of a sufficient length to reduce the impact of interannual variability. As such, a direct backward comparison of the metric grades in this paper to those of the CCMVal-2 model simulations (Butchart et al., 2010) is not possible. However, assuming that the change in the grades awarded to the UMUKCA-METO simulation (as re-calculated using the period 1980–2010) is representative of that for other chemistry-climate models, it is likely that the HadGEM3-ES free-running model performs better than the CCMVal-2 multi-model mean in 10 of the 14 metrics.

Particularly signficant improvements to the free-running model are that HadGEM3-ES no longer suffers from the large positive bias in stratospheric age of air or large warm bias in tropical tropopause temperature that were present in UMUKCA-METO (Morgenstern et al., 2010). More realistic stratospheric water vapour concentrations make HadGEM3-ES more suitable for accurately simulating stratospheric ozone concentrations (Hardiman et al., 2015). Issues do remain with the free-running model climatology, however. The seasonal cycle in extra-tropical winds and temperatures is found to be slightly weak in the model. This is most noticeable in the southern hemisphere polar vortex, which is too weak (by up to 6m s$^{-1}$) and therefore too warm (by up to 4K). There are also ongoing moderate biases in temperature, water vapour, ozone and upwelling mass flux in the tropics.

Metrics are split into those assessing mean climate and those assessing variability. The mean climate was found to be well simulated in both free-running and nudged versions of HadGEM3-ES with the notable exception of stratospheric transport, as diagnosed by the upwelling mass flux in the tropics. Vertical velocities are very noisy in reanalysis data (Butchart, 2014) and, therefore, cannot be nudged towards. As such, the diabatic component of stratospheric transport is difficult to constrain, even in nudged simulations. However, the variability in the nudged simulations was found to be significantly closer to the reanalysis than the variability in the free-running simulations. The nudged simulations showed grades above 0.9 for all variability metrics, except that diagnosing the accuracy of the quasi-biennial oscillation. In this case, the measure of variability used for the quasi-



biennial oscillation was found to make the metric too sensitive in general, demonstrating the use of nudged simulations for ensuring the robustness and reliability of metrics involving quantities that are directly nudged.

Comparison of the free-running model climatology to that of the nudged version shows that accurately simulated dynamics, specifically temperature and horizontal wind fields, do play a role in the spatial structure of the ozone hole. This structure is correct in both hemispheres in the nudged model. However, the high ozone biases that exist in the tropics and southern high latitudes of the free-running model persist also in the nudged model, and these are therefore not solely attributable to biases in the dynamical fields. Thus, despite the fact that the area of southern hemisphere Polar Stratospheric Clouds is correctly simulated in the nudged model, the ozone hole area is too small in both free-running and nudged models (an issue which is not unique to HadGEM3-ES, as shown by Figure 1 of Austin et al., 2010).

Ozone concentrations in the northern hemisphere winter are found to increase by as much as 15% following sudden strato-spheric warmings (SSWs). In free-running simulations, the SSWs are too weak, leading to a change in ozone concentrations that is also too weak (by as much as 5% of the ozone climatological concentrations). In the nudged model, both these biases disappear, demonstrating that errors in the re-distribution of ozone following a SSW are purely dynamical.

Tropical ozone concentrations are improved in the nudged simulations over those seen in the free-running model, but they are still biased high relative to observations, with these biases occurring in the tropical tropopause region. It is worth noting that both water vapour and ozone concentrations are not perfect in the nudged simulation, and significant biases in the simulated transport and chemistry (and potentially errors in e.g. convection, lightning emissions, and biomass burning emissions and their distribution; Stevenson et al., 2006) still exist in this model.

The fact that tropical upwelling and the stratospheric meridional circulation are found difficult to constrain and, indeed, are found to be worse in the nudged simulations than in the free-running simulations, means that ozone fluxes, in particular from the stratosphere to the troposphere, are not well constrained in the nudged model either, with obvious implications for the simulated extratropical tropospheric ozone budget. Again this issue is not unique to HadGEM3-ES – even the ERA-Interim reanalysis shows ozone fluxes from the stratosphere to the troposphere with only around half the value inferred from observations.

In summary, biases in transport and ozone remain in the nudged simulations, demonstrating that these biases are not solely due to the model dynamics. Nevertheless, HadGEM3-ES is found to have good climatology and variability in basic meteoro-logical fields, and a realistic simulation of stratospheric ozone loss. HadGEM3-ES represents a significant improvement over its predecessor, UMUKCA-METO, and compares favourably with other previous chemistry-climate models.

**Code and data availability**

Due to intellectual property right restrictions, we cannot provide either the source code or documentation papers for the UM. The Met Office Unified Model is available for use under licence. A number of research organisations and national meteoro-logical services use the UM in collaboration with the Met Office to undertake basic atmospheric process research, produce forecasts, develop the UM code and build and evaluate Earth system models. For further information on how to apply for a licence see http://www.metoffice.gov.uk/research/modelling-systems/unified-model. JULES is available under licence free of



charge. For further information on how to gain permission to use JULES for research purposes see https://jules.jchmr.org/software-and-documentation.

The model code for NEMO v3.4 is available from the NEMO website (www.nemo-ocean.eu). On registering, individuals can access the code using the open source subversion software (http://subversion.apache.org/). The revision number of the base NEMO code used for this paper is 3309. The model code for CICE is freely available from the United States Los Alamos National Laboratory (http://oceans11.lanl.gov/trac/CICE/wiki/SourceCode), again using subversion. The revision number for the version used for this paper is 430.

The data will be submitted to the British Atmospheric Data Center (BADC) database for the CCMI project.

*Author contributions.* SCH wrote sections 1, 3 and 4 of the manuscript and produced the figures. FMO'C wrote section 2 of the manuscript. SCH, NB, and FMO'C contributed to running model integrations and to discussion on the structure and content of the manuscript. STR processed the chemistry and aerosol emissions datasets used in model integrations.

*Acknowledgements.* The work of SCH, NB, FMO'C and STR was supported by the Joint UK BEIS/Defra Met Office Hadley Centre Climate Programme (GA01101). SCH and NB were also supported by the European Commission's 7th Framework Programme, under Grant Agreement number 603557, StratoClim project. FMO'C also acknowledges additional funding received from the Horizon 2020 European Union's Framework Programme for Research and Innovation "Coordinated Research in Earth Systems and Climate: Experiments, kNowledge, Dissemination and Outreach (CRESCENDO)" project under Grant Agreement No 641816. The authors would like to acknowledge Jeff Knight for processing the sea surface temperature and sea ice datasets used in the model integrations. ERA-Interim data used in this study were provided by ECMWF. Met Office CCMVal-2 data used in this study is stored at the British Atmospheric Data Centre (BADC). We acknowledge use of the MONSooN high performance computing system, a collaborative facility supplied under the Joint Weather and Climate Research Programme, a strategic partnership between the Met Office and the Natural Environment Research Council.



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





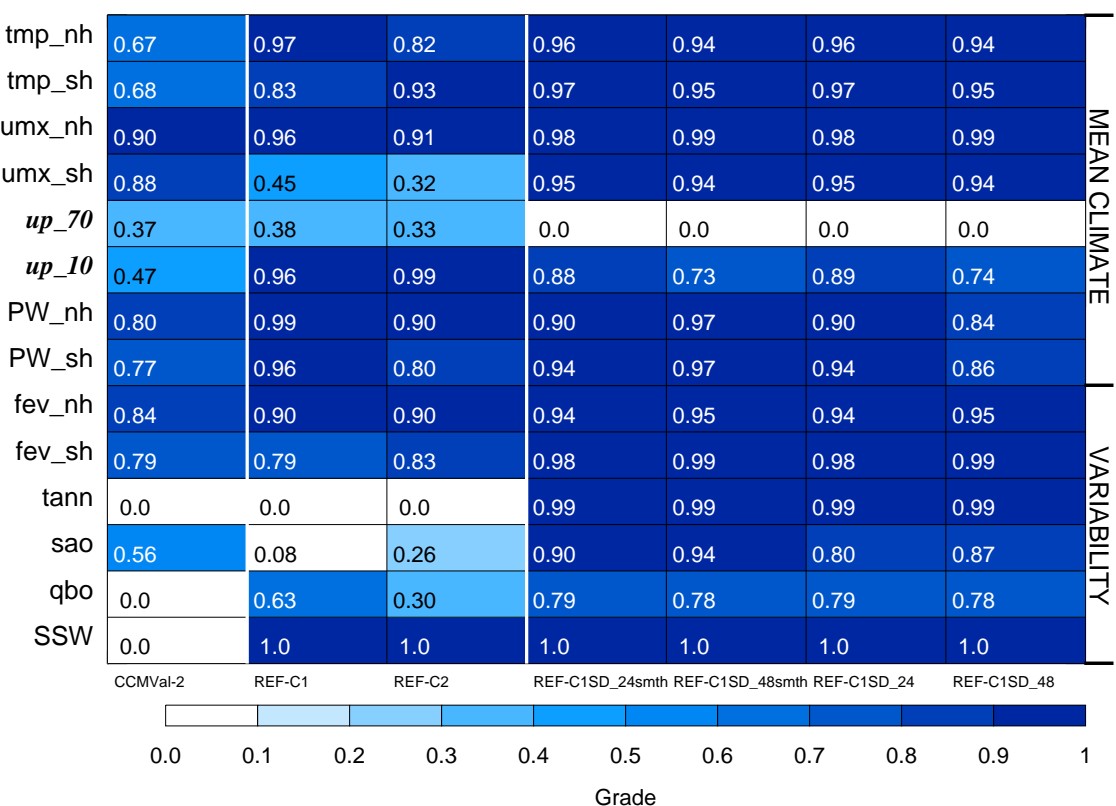

**Figure 1.** Metrics of dynamical fields and processes (see Table 2). Bold italic font indicates metrics which are not directly constrained in the nudged simulations. For details of model simulations see Table 1 (where "24smth" corresponds to "24hr, smoothed" etc.).





**Figure 2.** (a) Zonal mean annual mean temperature for the REF-C1 simulation, (b) As (a) but differences between the REF-C1 simulation and ERA-Interim, (c) Zonal mean zonal wind, for December-January-February (northern hemisphere) and June-July-August (southern hemisphere), for the REF-C1 simulation, (d) As (c) but differences between the REF-C1 simulation and ERA-Interim. The years 1980–2010 are used.





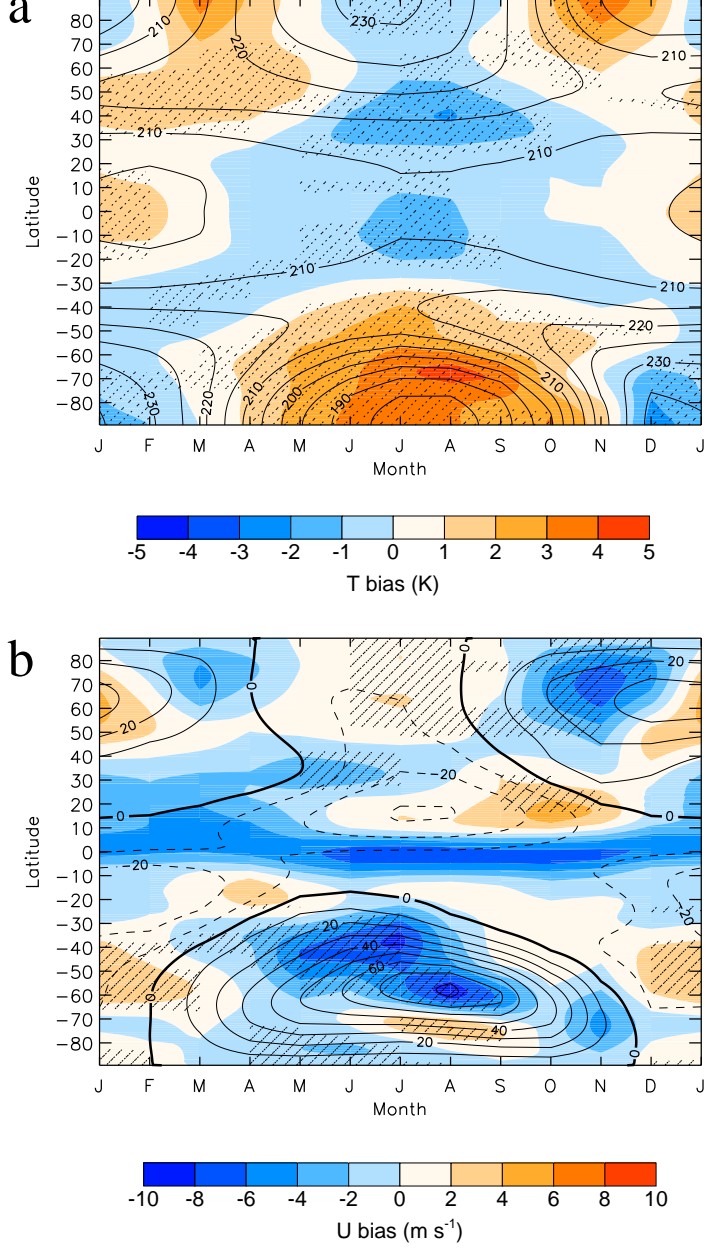

**Figure 3.** Biases in the climatological seasonal cycle of the REF-C1 simulation, relative to ERA-Interim, for zonal mean (a) Temperature (50hPa) and (b) Zonal wind (10hPa). Black contours show ERA-I values, with contour intervals of 5K and 10m s$^{-1}$ respectively, and coloured shading shows the bias (REF-C1 minus ERA-I), with contour intervals 1K and 2m s$^{-1}$ respectively. Stippling shows regions where the bias is statistically significant at the 95% level as calculated using a T-test. Tick marks indicate the middle of each month.





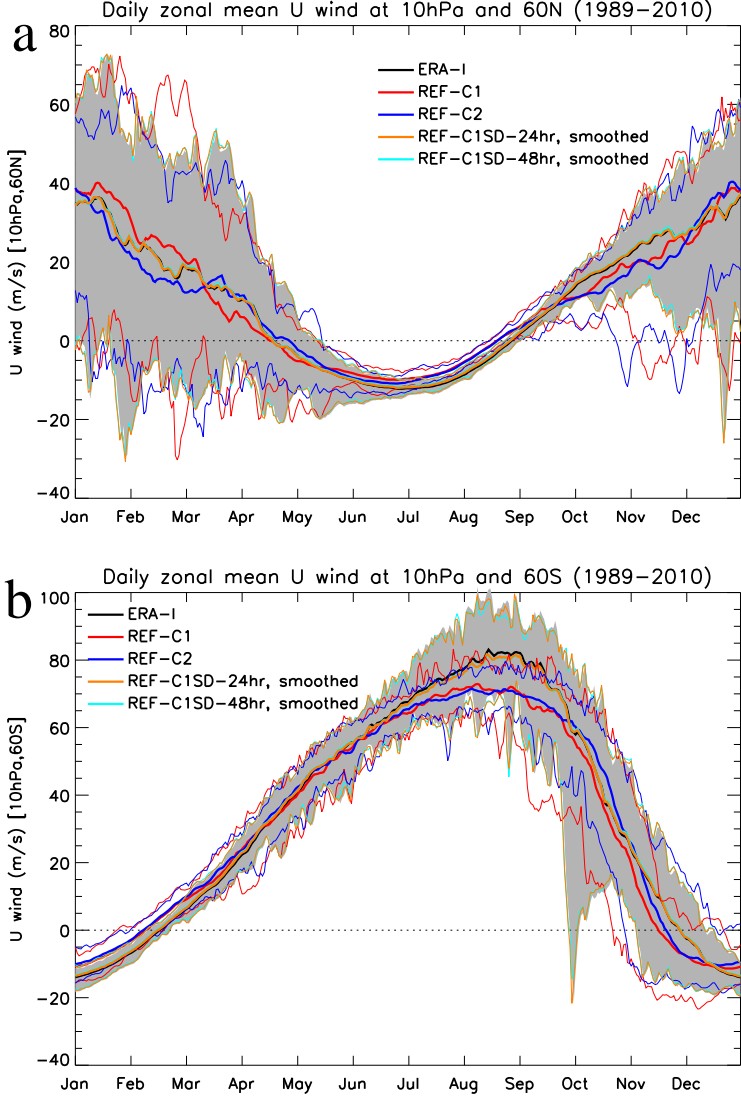

**Figure 4.** Polar vortex variability for (a) Northern hemisphere and (b) Southern hemisphere. Thick solid lines show mean values, and maximum and minimum values are shown by thin solid lines for the model simulations and shading for ERA-I, over the years 1989–2010. Tick marks indicate the middle of each month.





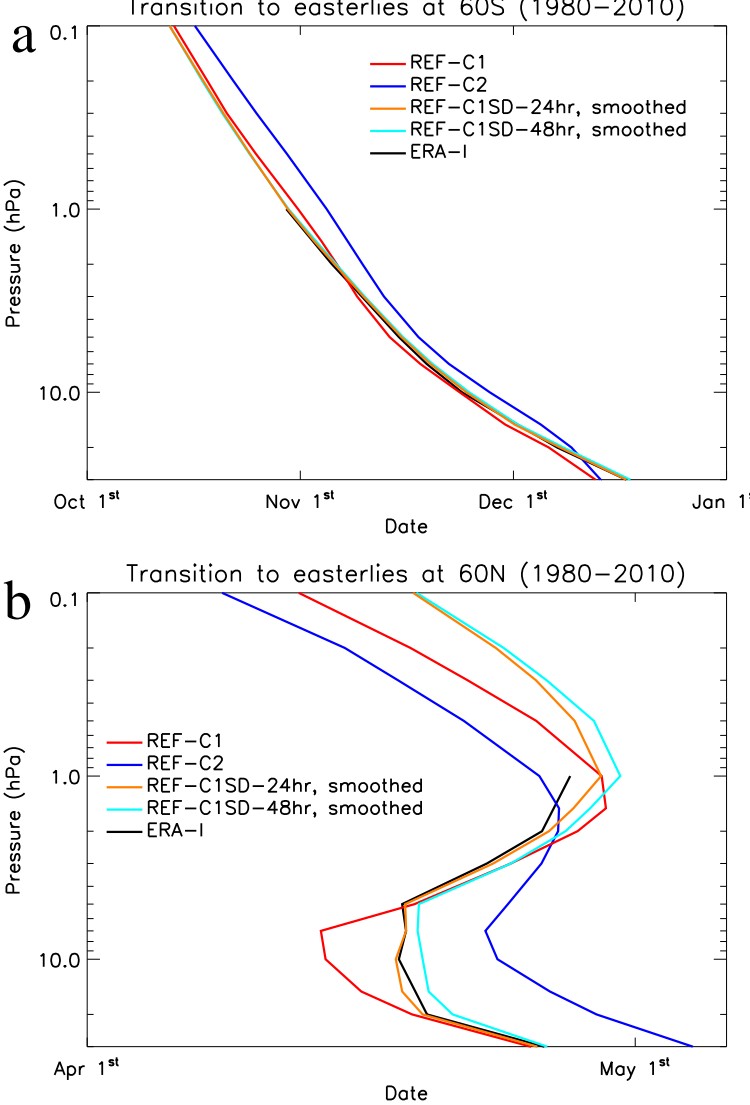

**Figure 5.** Polar vortex final warming times, as defined by the final transition from eastward to westward of the zonal mean zonal wind at 60°, for (a) the southern hemisphere and (b) the northern hemisphere. Climatologies for the years 1980–2010 are shown.






**Figure 6.** (a) Average daily October Nitric Acid Trihydrate (NAT) PSC area in the southern hemisphere, defined as the area poleward of 60°S with daily mean temperatures below 195K. (b) Accumulated daily PSC area in the northern hemisphere, defined as the area poleward of 60°N with daily mean temperature below 195K. (c) Minimum 50hPa daily mean temperature in the region 60°S–90°S. (d) Minimum 50hPa daily mean temperature in the region 60°N–90°N. Thick and thin lines, and shading, in panels (c) and (d) are as in Figure 4. All panels are averaged over years 1989-2009. *Note that temperature is used as a proxy for PSC area here, and thus these are estimates of the PSC area seen by the interactive chemistry.*





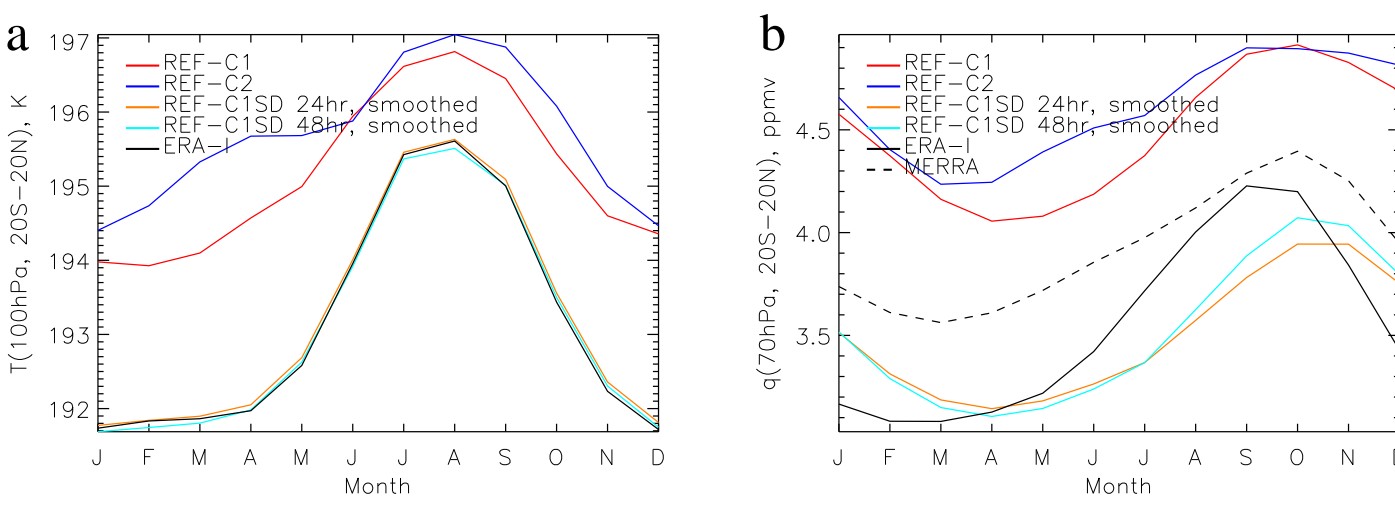

**Figure 7.** Tropical (20°S–20°N) seasonal cycle in (a) temperature (T) and (b) water vapour (q), averaged over the years 1980–1999, as compared to ERA-Interim reanalysis (for T), and ERA-I and MERRA reanalyses (for q). Tick marks indicate the middle of each month.

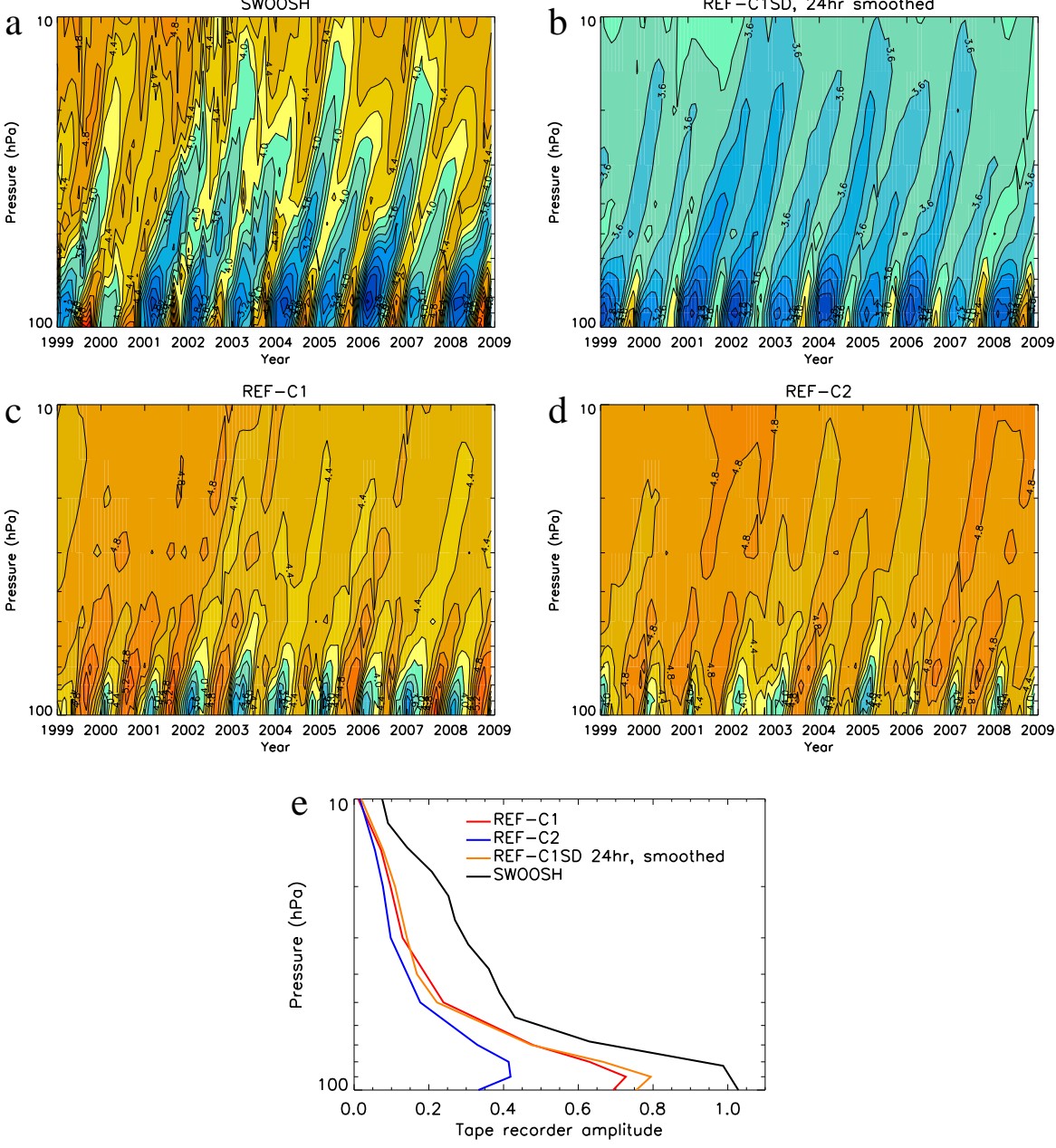

**Figure 8.** Tropical "tape recorder" signal, q (ppmv) averaged 10°S–10°N, for (a) SWOOSH data, and the (b) REF-C1SD 24hr smoothed, (c) REF-C1 and (d) REF-C2 simulations. (e) Amplitude of tape-recorder calculated, at each height, as the amplitude of the Fourier harmonic corresponding to the annual cycle.





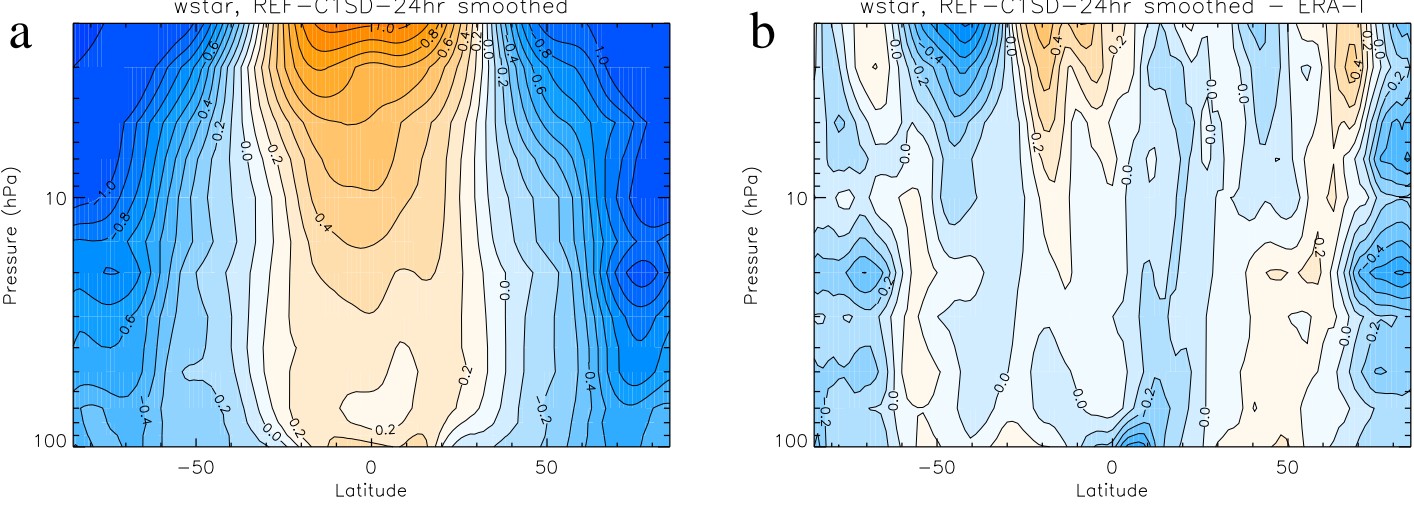

**Figure 9.** Zonal mean annual mean climatologies in residual vertical velocity for (a) REF-C1SD (nudged simulation) and (b) Differences between the REF-C1SD simulation and ERA-Interim. The years 1989–2009 are used. Unlike temperature and zonal wind, the biases in residual vertical velocity are *not* negligible for the nudged simulations (see text for details).







**Figure 10.** (a) Residual vertical velocity at 70hPa (1989–2009), and tropical mass upwelling through 70hPa for (b) annual mean, (c) December-January-February, and (d) June-July-August, as calculated for free-running simulations, nudged simulations and ERA-Interim. Mass upwelling in (b) is calculated using seasonal means as in Butchart et al. (2010), such that the annual means plotted above the x-tick marks refer to Dec–Nov means.





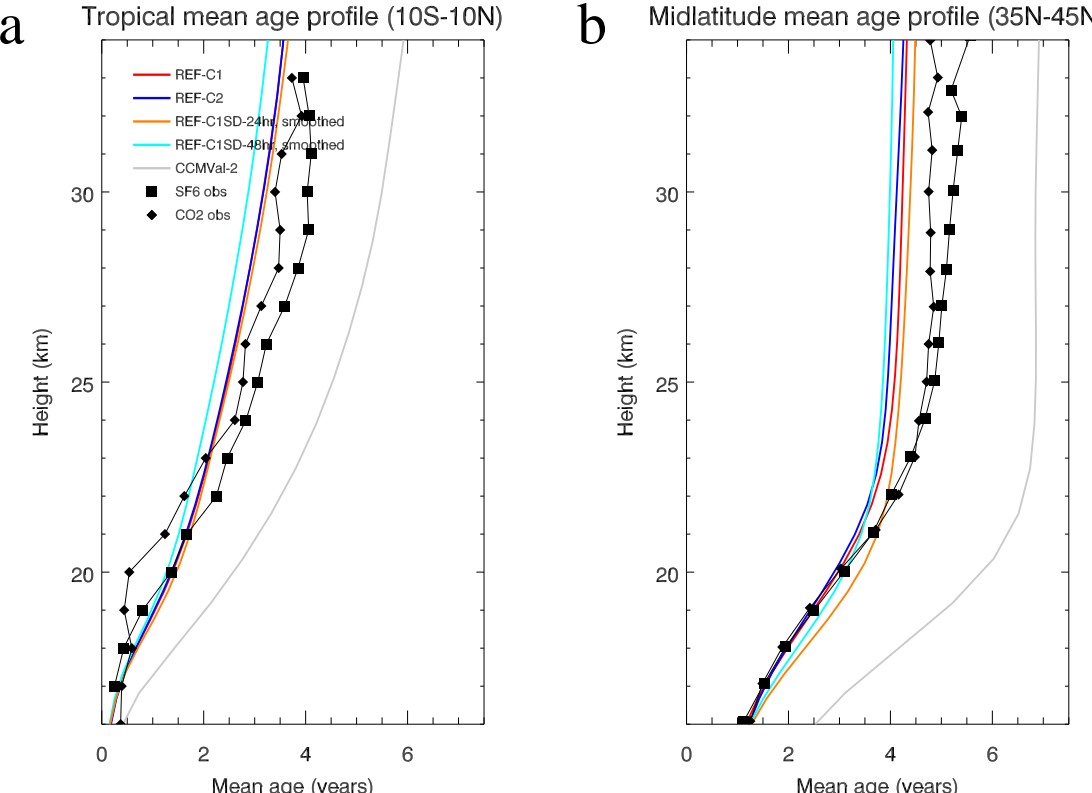

**Figure 11.** Stratospheric age of air (1990–2010) in the (a) Tropics (observations from Andrews et al., 2001) and (b) Northern Hemisphere mid-latitudes (observations from Engel et al., 2009). The period 1990–2010 was chosen for CCM-I model simulations to allow for age of air to adjust during the first 10 years of the nudged simulations. The period 1980–2000 was used for the CCMVal-2 model simulation (historical period only). The exact period chosen makes very little difference to the diagnosed age of air (not shown).





**Figure 12.** Column ozone: (a) annual mean near-global (60°S–60°N), (b) annual mean tropics (20°S-20°N), (c) northern hemisphere March (60°N-90°N), and (d) southern hemisphere October (60°S-90°S).





**Figure 13.** Stratosphere-Troposphere-Exchange of ozone for (a) annual mean, (b) December-January-February, and (c) June-July-August. This flux of ozone across the tropopause is calculated using monthly mean residual vertical velocity and ozone mass mixing ratio, following Hegglin and Shepherd (2009). The tropopause is here defined as the 100hPa surface equatorward of 50° and the 200hPa surface poleward of 50°.



**Figure 14.** Climatological column ozone during October in the southern hemisphere for (a) REF-C1, (b) REF-C2, (c) REF-C1SD-24hr (smoothed), and (d) TOMS. (e) Ozone hole, defined as the 220DU contour. White contour in (a), (b) and (c) shows TOMS 220DU contour. Ozone concentrations in REF-C1SD are still biased high, but the ozone hole has the correct shape. Years 1997–2002 are used in all cases.





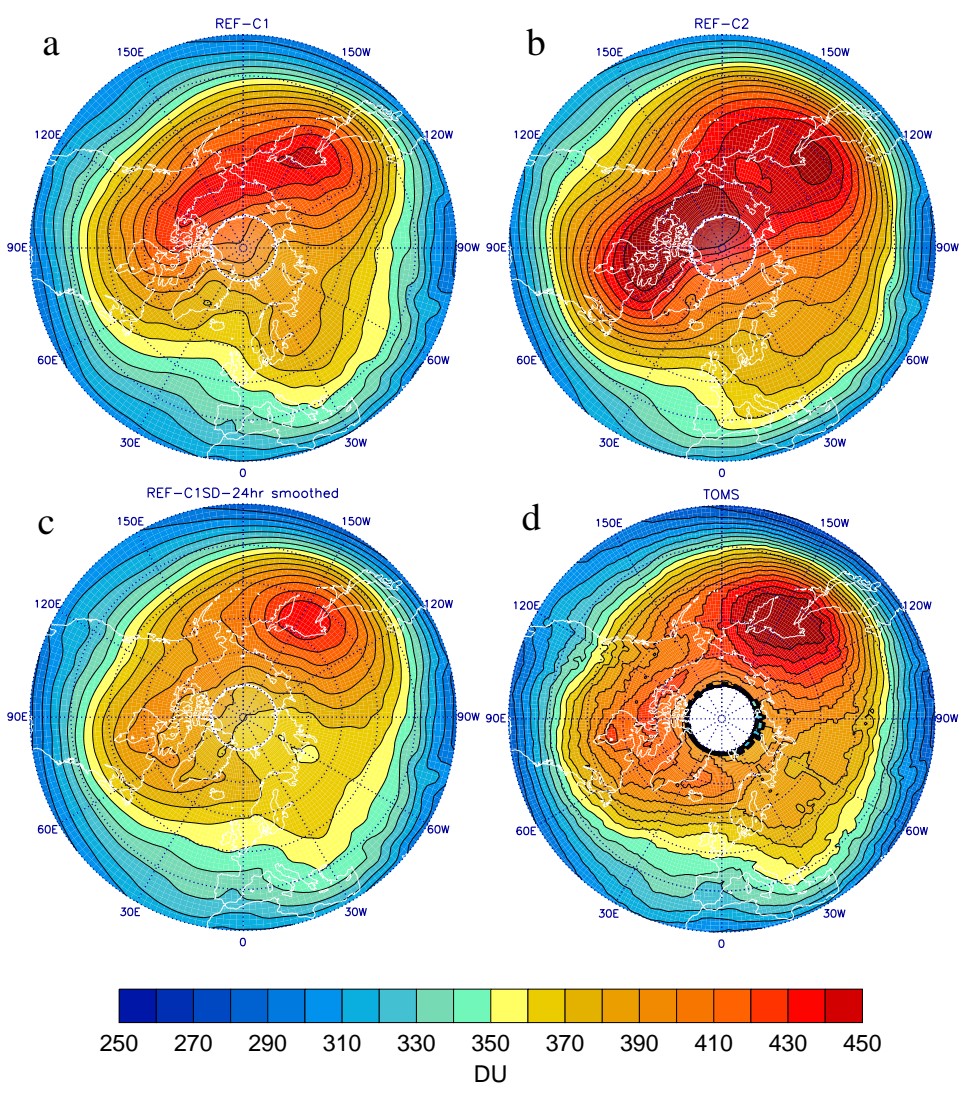

**Figure 15.** As Figure 14 panels (a)–(d), but for climatological column ozone in northern hemisphere March.



**Figure 16.** Anomalies, averaged over the 30 days following a stratospheric sudden warming, in (a, b, c) Ozone volume mixing ratio (ppmv), (d, e, f) Ozone, as percentage of climatological values, and (g, h, i) temperature (K), for ERA-Interim, 24hr nudged simulation and free-running REF-C1 simulation. Stippling shows regions where the anomalies are statistically significantly different from zero, with 95% confidence, as calculated using a T-test.





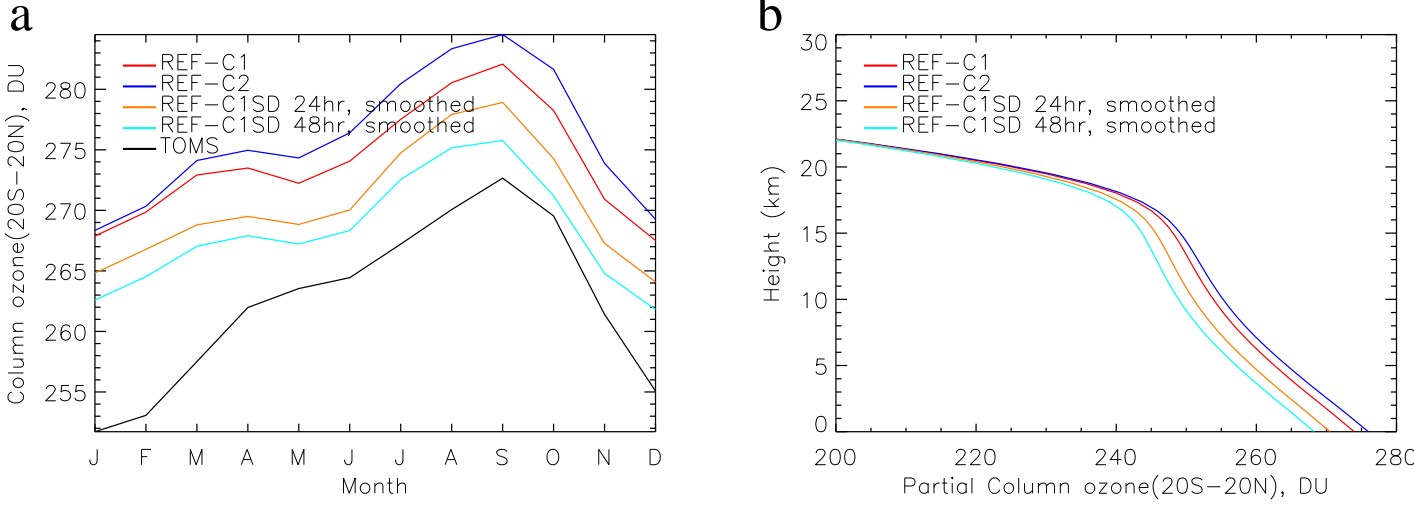

**Figure 17.** (a) Seasonal cycle in tropical column ozone, averaged over the years 1980–1999, as compared to TOMS satellite data. Tick marks indicate the middle of each month. (b) Vertical profile of partial column ozone, integrated downwards from the top of the model.



**Table 1.** Model simulations

| Name | Time period | Coupled Ocean? | Nudging time scale | Smoothing? |
|---|---|---|---|---|
| REF-C1 | 1960–2010 | No | N/A | N/A |
| REF-C2 | 1960–2100 | Yes | N/A | N/A |
| REF-C1SD-24hr | 1980–2010 | No | 24 hours | No |
| REF-C1SD-48hr | 1980–2010 | No | 48 hours | No |
| REF-C1SD-24hr, smoothed | 1980–2010 | No | 24 hours | Yes |
| REF-C1SD-48hr, smoothed | 1980–2010 | No | 48 hours | Yes |
| CCMVal-2 (UMUKCA-METO) | 1960–2005 | No | N/A | N/A |


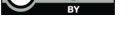

**Table 2.** Metrics

| Name | Description |
| --- | --- |
| *Mean Climate* | |
| tmp_nh | 60-90°N December-January-February temperatures at 50hPa |
| tmp_sh | 60-90°S September-October-November temperatures at 50hPa |
| umx_nh | Maximum northern hemisphere eastward wind in December-January-February at 10hPa |
| umx_sh | Maximum southern hemisphere eastward wind in June-July-August at 10hPa |
| up_70 | Tropical upwelling mass flux at 70hPa |
| up_10 | Tropical upwelling mass flux at 10hPa |
| PW_nh | Slope of the regression of the February and March 50hPa temperatures 60-90°N on the 100hPa January and February heat flux 40-80°N |
| PW_sh | Slope of the regression of the August and September 50hPa temperatures 60-90°S on the 100hPa July and August heat flux 40-80°N |
| *Variability* | |
| fev_nh | Amplitude of the leading mode of variability (EOF) of the 50hPa zonal-mean zonal wind for the northern hemisphere, poleward of 45°. EOFs are scaled to have the same standard deviation as the original data. |
| fev_sh | Amplitude of the leading mode of variability (EOF) of the 50hPa zonal-mean zonal wind for the southern hemisphere, poleward of 45°. EOFs are scaled to have the same standard deviation as the original data. |
| tann | Amplitude of the annual cycle at 2hPa in the zonal-mean zonal wind, 10°S-10°N |
| sao | Amplitude of the semi-annual oscillation at 1hPa in the zonal-mean zonal wind, 10°S-10°N |
| qbo | Amplitude of the quasi-biennial oscillation at 20hPa in the zonal-mean zonal wind, 10°S-10°N |
| SSW | Frequency per year of major sudden stratospheric warmings, defined using reversal of the zonal-mean zonal wind at 10hPa, 60°N |