# Peer review of "The Met Office HadGEM3-ES Chemistry-Climate Model: Evaluation of stratospheric dynamics and its impact on ozone"

_Geoscientific Model Development, 2016_

## Referee Comment (RC1) · Anonymous Referee #1 · 24 Dec 2016

General Comments This work evaluates the stratospheric dynamics and impact on ozone of the Met Office HadGEM3-ES chemistry-climate model. The authors have done an excellent job of describing the new version (compared to the previous CCM-Val2 version designated as UMUKCA-METO). They have examined 14 dynamical metrics and graded the model in the manner of Waugh and Erying 2008. Overall I fine this study appropriate for GMD and recommend it for publication. I some specific comments below that would improve the current draft.

Specific Comments

Since it is not stated, I assume that the REFC1 and REFC2 simulations only use one ensemble member, correct? This will limit what you can say about variability. For

example, your comments on Page 8, lines 6-8.

Page 3, line 27 In most (if not all publication), the Chemistry-Climate Model Initiative is designated as CCMI, not CCM-I.

Page 4, line 5. The authors state the horizontal winds and temperature are nudged. Question, many groups that use a specified dynamics approach also nudge surface pressure. I am assuming you don't do this because you only nudge over the 2.5km-51km range, therefore not nudging the surface region? Could you give a few more detail on why you made this choice?

Also, how do you transition to the free running version above 51km?

Page 6 discussion of Figure1. One very minor suggestion would be to add column numbers at the top of Figure 1 since you are specifically identifying columns in the text. It will make it a bit easier for the reader to quickly follow the discussion.

Page 6, lines 26-30, and Figure1 (QBO nudging). I am also surprised that the SD version grade in Figure1 is only 0.8. Your explanation makes sense; however, I have one clarifying question. The reanalysis implicitly has a representation of the tropical zonal winds (QBO) based on observation. Therefore, when you run in SD are you also nudging the model explicitly with a relaxation to Singapore winds (similar to what is done in a REFC1 simulation)? This could cause issues if the nudging is essentially done twice.

Page 8, lines 29-30. Please give a brief summary of the PSC approach (i.e., do you represent NAT, water-ice, and supercooled ternary solution (STS) PSCs?).

Discussion of Figure 3 (lat/time T at 50hPa), Figure 6a (Oct polar cap avg PSC area, 50hPa?), and Figure 12d (SH column ozone). I can understand that the free running model may not give good ozone depletion, but why doesn't the SD version? In SD you have temperatures and vortex area that are well represented. So why is the total column ozone ∼50DU higher than observations? Doesn't this say something about

the PSC/heterogeneous chemistry parameterization in the model? Or does this have something to do with the advection routine being too diffusive?

Page 10 and the discussion of Figure 8b. (SD version) You state that the "tape-recorder signal appears more coherent far higher in the stratosphere in the nudged simulations. However, Figure 8(e) shows that this is not due to the amplitude of the annual cycle harmonic." I'm a bit confused by this statement, since, the "dry phase" of the tape recorder seems to represent the SWOOSH data well at the entry level and the propagation upward. This does not seem to be the case for the "wet phase". Does this say something about the robustness of the models' microphysical parameterization of ice (i.e., too much dehydration)?
* * *

---

## Referee Comment (RC2) · Anonymous Referee #2 · 25 Dec 2016

This paper presents an evaluation of stratospheric dynamics and its impact on ozone in the UKMO HadGEM3-ES model. The authors make comparisons between the free-running and the nudged versions, mainly focusing on stratospheric dynamical properties and total ozone columns, and conclude that the dynamical processes are better presented in the nudged version, although there are still significant biases in simulating stratospheric transport, water vapour, and ozone columns. By comparing the metrics of some dynamical processes that are relevant to simulating stratospheric ozone, the authors also conclude that the present model version is significantly improved compared the previous model version that was used in the CCMVal2 inter-model comparison, for the majority of the tested metrics.

[Figure]

Overall, the paper is well written with sufficient detail; it will make a valuable contribution to understanding how chemistry-climate model (CCM) biases (which are mainly dynamical) impact simulated ozone columns, and can be used as a benchmark for future UKMO CCM development. The paper is appropriate for publication in GMD, after some revisions (see specific comments below). I also encourage the authors to consider the following suggestions.

Suggestions:

Although the paper's structure is clear, I think "Section 3.1 Metrics" would be better placed after the detailed comparisons of dynamical properties and ozone. Moreover, most metrics calculated are not referred to in the following comparisons of dynamical properties and their impact on ozone. My suggestion would be to split the "Results" section into two sections, i.e. "evaluation of stratospheric dynamics and ozone", and "Quantitative assessment, i.e., metrics".

More could be made of the differences in model behaviour between REF-C1 and REF-C2. REF-C1 is usually closer to observations than REF-C2, as expected.

Specific comments:

1) "Ozone concentrations" appear throughout the paper, but the authors only show total column ozone (TCO). So the authors should replace all "ozone concentrations" with TCO. They are not the same, therefore should not be mixed.

2) In the abstract, the last sentence says that "... that the nudged models still remain far from perfect": Could you elaborate in which sense these models are "far from perfect"? I suggest to re-phrase this statement, and point out any potential problems in applying nudging techniques. It feels like an empty statement to me.

3) P3L22: the previous version used in CCMVal2 did have interactive lightning NOx emissions and interactive wet deposition although for a much more limited range of species. Dry deposition used offline tabulated deposition velocities (Morgenstern et

al., 2009). Please correct.

4) P6, paragraph 3, you state that nudged simulations do not perform well in metrics of "tropical upwelling and QBO"; could you elaborate on any inconsistencies in treating model's dynamics in nudging and their impact on some simulated model properties? You may want to mention the idea that wind fields used for nudging may not satisfy the continuity equation, which will negatively impact vertical velocity fields.

5) Section 3.3.1 "Extratropics" only covers high-latitude aspects. I suggest to either re-title the section to "High latitudes" or give some coverage to mid-latitude aspects.

6) P11L24: Replace "ozone depletion" with total column ozone (TCO, the standard notation). You're not actually quantifying ozone depletion, just total columns. Also L25: Replace "column ozone concentrations" with TCO.

7) P12L12: That is technically correct, but imposing zonally invariant ozone would not improve the situation. Rather than imposing zonally invariant ozone (which would be inconsistent with best understanding of the ozone distribution), would it be more effective to work on the model to improve the factors that influence the phase of these planetary waves, such as orographic forcing? The discussion of how to impose ozone in models that cannot get the phase of the waves correct strikes me as somewhat missing the point.

8) L12L25: I noticed that there is a negative trend in tropical ozone in all simulations, but there does not appear to be much trend in the observations. Please comment on this.

9) P13L2: It is true that convection, lightning emissions, and BB could impact tropospheric ozone, but they are unlikely the main cause for the 10 DU bias in TCOs here. Actually figure 17b suggest that it's mainly the tropopause height whose variations give you differences in TCO between the simulations. In the troposphere, to partial columns go in parallel (implying there is no significant difference in tropical tropospheric ozone

between the simulations). I think your suggestion that tropospheric processes cause this high bias is insufficiently supported by your findings. If this were purely a tropospheric problem, 10 DU would likely amount to an unrealistic 50% error in tropical tropospheric ozone. More likely, it is due mainly to an error in the placement of the tropical tropopause, which you could establish.

10) P13L21: Morgenstern et al. (2009) is a more appropriate reference here. This problem was not specifically addressed in Morgenstern et al. (2010).

11) P14L7 (cf. Figure 14): That's surprising, considering there should be a close correspondence between the size of the polar vortex, as defined by a transport barrier, and the ozone hole (which is bounded by that transport barrier). If despite nudging these two still differ, could it be that the reanalyses are insufficiently constrained by observations during winter/spring over Antarctica? Please elaborate on the role of the transport barrier in this.

12) P14L17: Your analysis does not imply errors in any of these processes. To make such a statement, you would have had to compare tropospheric ozone against observations. See above on the role of the tropopause height.

13) P14L27: You did not directly compare this model version against other models, so I suggest to remove this half-sentence.

---

## Author Comment (AC1) · 22 Feb 2017

**Response to reviewers comments on "The Met Office HadGEM3-ES Chemistry-Climate Model: Evaluation of stratospheric dynamics and its impact on ozone"**

A bug in the mid 1970s of the original REF-C1 simulation has been discovered since the submission of this manuscript, and it was not known whether this bug affected the period 1980–2010 of the simulation. As such, the REF-C1 simulation has been redone and all figures in the paper have been reproduced to use this new, bug free, REF-C1 simulation. Although there are minor differences to some numbers quoted in the text, use of this new simulation has made no difference to any of our conclusions, with one exception. Stratospheric Sudden Warmings (SSWs) in the new REF-C1 are found to be just as well simulated as those in the nudged simulation. As such, we have re-written the conclusions surrounding the SSW results. All the changes made since the original submission are included in the track changes document, along with changes due to the reviewers comments below.

The authors thank the reviewers for their detailed comments on the manuscript. Our responses to these comments follow.

**Anonymous Referee 1**

**General Comments**

**This work evaluates the stratospheric dynamics and impact on ozone of the Met Office HadGEM3-ES chemistry-climate model. The authors have done an excellent job of describing the new version (compared to the previous CCMVal2 version designated as UMUKCA-METO). They have examined 14 dynamical metrics and graded the model in the manner of Waugh and Eyring 2008. Overall I fine this study appropriate for GMD and recommend it for publication. I some specific comments below that would improve the current draft.**

**Specific Comments**

**Since it is not stated, I assume that the REFC1 and REFC2 simulations only use one ensemble member, correct? This will limit what you can say about variability. For example, your comments on Page 8, lines 6-8.**

A single ensemble member for each of the REF-C1 and REF-C2 simulations is documented and studied in this paper. A sentence to clarify this has been

added to Section 2. The comment on Page 8, lines 6–8, refers to the interannual variability over the 30 years of this single ensemble member. The word "interannual" has been inserted in the text to make this clear. However, it is true that we have run extra ensemble members, which are not documented in this paper nor intended for upload to the CCMI database, which we have used for information in the text surrounding Figure 6(b).

*Changes in manuscript: Inserted the text "a single ensemble member for each of" in paragraph 4 of Section 2 (P3L32). Inserted the word "interannual" in first paragraph of Section 3.2.1 (P8L22).*

**Page 3, line 27. In most (if not all) publication, the Chemistry-Climate Model Initiative is designated as CCMI, not CCM-I.**

Thank you for pointing this out. This has now been changed.

*Changes in manuscript: CCM-I globally replaced with CCMI.*

**Page 4, line 5. The authors state the horizontal winds and temperature are nudged. Question, many groups that use a specified dynamics approach also nudge surface pressure. I am assuming you don't do this because you only nudge over the 2.5km-51km range, therefore not nudging the surface region? Could you give a few more detail on why you made this choice? Also, how do you transition to the free running version above 51km?**

The original documentation for the nudging is Telford et al. (2008), as referenced in the manuscript. As the reviewer correctly points out, surface pressure is not nudged, although Telford et al. (2008) show that it is fairly accurately simulated. The reasons that surface pressure is not nudged are as follows. The Met Office model has a non-hydrostatic terrain following dynamical core, and surface pressure is not a model prognostic. Further, the difference in horizontal resolution between the model and the reanalysis data would lead to a mismatch in the details of the orography. Nudging is smoothly increased over the 2 model levels above a height of 2.5km, and smoothly decreased over the 2 model levels below a height of 51km. Thus the nudging is not suddenly terminated in the vertical at 51km. The model is free-running above 51km, as the reviewer states.

*Changes in manuscript: The following text has been added to paragraph 6 of Section 2 (P4L11–14): "Nudging is applied over the vertical range 2.5km – 51km, and is smoothly increased/decreased over two model levels at the bottom/top of this vertical range. Surface pressure is not nudged, since HadGEM3-ES has a non-hydrostatic terrain following dynamical core in which surface pressure is not a prognostic and, further, the difference in horizontal*

*resolution between the model and the reanalysis data would lead to a mismatch in details of the orography."*

**Page 6 discussion of Figure 1. One very minor suggestion would be to add column numbers at the top of Figure 1 since you are specifically identifying columns in the text. It will make it a bit easier for the reader to quickly follow the discussion.**

Change made.

*Changes in manuscript: Column numbers have been added to Figure 1. The text "Column numbers are printed above each column, and the model simulation is printed below each column." has been added to the Figure 1 caption.*

**Page 6, lines 26-30, and Figure 1 (QBO nudging). I am also surprised that the SD version grade in Figure 1 is only 0.8. Your explanation makes sense; however, I have one clarifying question. The reanalysis implicitly has a representation of the tropical zonal winds (QBO) based on observation. Therefore, when you run in SD are you also nudging the model explicitly with a relaxation to Singapore winds (similar to what is done in a REFC1 simulation)? This could cause issues if the nudging is essentially done twice.**

The QBO is internally generated in HadGEM3-ES, and as such is not nudged in any way in the free-running REF-C1 simulation. No nudging towards Singapore winds occurs in any of the HadGEM3-ES simulations. This has been made clearer in the manuscript.

*Changes in manuscript: The text in paragraph 9 of Section 3.1 (P7L3–5) has been modified to read "Although the QBO is internally generated in the free-running REF-C1 and REF-C2 simulations, the QBO metric depends only on zonal wind which is directly nudged in the REF-C1SD simulations."*

**Page 8, lines 29-30. Please give a brief summary of the PSC approach (i.e., do you represent NAT, water-ice, and supercooled ternary solution (STS) PSCs?). Discussion of Figure 3 (lat/time T at 50hPa), Figure 6a (Oct polar cap avg PSC area, 50hPa?), and Figure 12d (SH column ozone). I can understand that the free running model may not give good ozone depletion, but why doesn't the SD version? In SD you have temperatures and vortex area that are well represented. So why is the total column ozone ~50DU higher than observations? Doesn't this say something about the PSC/heterogeneous chemistry parameterization in the model? Or does this have something to do with the advection routine being too diffusive?**

A summary of the model PSC approach has now been added to Section 2 of the manuscript. The amount of depletion, from 1980 to 2000, simulated in total column ozone in October in the southern high latitudes agrees with the observations, but the total column ozone values are biased high by around 40DU. Whilst the similarity between free-running and nudged models allows restriction of the causes of this ozone bias to the model transport and chemistry schemes (as already noted in the manuscript), it is difficult to say anything more explicit than this. Indeed, ozone is biased high outside of high latitudes also, not just in PSC regions. However, Figure 3-11(c) from Chapter 3 of the 2010 WMO Ozone assessment report shows that a high bias of 40DU in this diagnostic is within the 95% prediction interval of the CCMVal-2 model simulations. This is now mentioned in the text.

*Changes in manuscript: The following text has been added to paragraph 3 of Section 2 (P3L21–25): "Details of the simulation of Polar Stratospheric Clouds (PSCs) are given in section 2 of Morgenstern et al. (2009) and section 2 of Chipperfield et al. (1998). Above the nitric acid trihydrate (NAT) point (195K), reactions occur on liquid sulfuric acid aerosols. Below this temperature the model forms solid NAT particles, and then below the ice point (188K) the model forms ice particles. There is no representation of supercooled ternary solutions.". The following text has been added to paragraph 1 of Section 3.3.1 (P12L14–15): "Figure 3-11(c) from Chapter 3 of WMO (2011) shows this bias to be within the 95% prediction interval of the CCMVal-2 model simulations.".*

**Page 10 and the discussion of Figure 8b. (SD version) You state that the "tape-recorder signal appears more coherent far higher in the stratosphere in the nudged simulations. However, Figure 8(e) shows that this is not due to the amplitude of the annual cycle harmonic." I'm a bit confused by this statement, since, the "dry phase" of the tape recorder seems to represent the SWOOSH data well at the entry level and the propagation upward. This does not seem to be the case for the "wet phase". Does this say something about the robustness of the models' microphysical parameterization of ice (i.e., too much dehydration)?**

The inclusion of Figure 8(e) was largely due to the fact that upward propagation cannot really be determined by eye-balling Figures 8(a)-8(d). The contour intervals chosen in these panels (regardless of their values) will make some features stand out more than others. However, it is the case that there is an overall dry bias in the nudged simulation. Figure 7(b) shows this to be around 0.5ppmv at 70hPa, relative to MERRA. Figure 7 of Hardiman et al. (2015) shows that, in more recent versions of the Met Office model, improvements to the ice microphysics scheme does lead to an increase in water vapour in the tropical tropopause layer of around this magnitude. This point has now been added to the discussion of Figure 7(b) in the current manuscript.

*Changes in manuscript: The relevant text in paragraph 1 of Section 3.2.2 (P10L8–12) has been modified to read "However, note that just nudging the temperatures and horizontal winds is not enough to remove any bias in water vapour concentrations (see also Hardiman et al., 2015). These are too low relative to the MERRA reanalysis by around 0.5ppmv (Figure 7(b)), although Figure 7 of Hardiman et al. (2015) suggests that improvements to the ice microphysics scheme in more recent versions of HadGEM may account for a significant fraction of this bias.".*

**Anonymous Referee 2**

This paper presents an evaluation of stratospheric dynamics and its impact on ozone in the UKMO HadGEM3-ES model. The authors make comparisons between the free-running and the nudged versions, mainly focusing on stratospheric dynamical properties and total ozone columns, and conclude that the dynamical processes are better presented in the nudged version, although there are still significant biases in simulating stratospheric transport, water vapour, and ozone columns. By comparing the metrics of some dynamical processes that are relevant to simulating stratospheric ozone, the authors also conclude that the present model version is significantly improved compared the previous model version that was used in the CCMVal2 inter-model comparison, for the majority of the tested metrics.

Overall, the paper is well written with sufficient detail; it will make a valuable contribution to understanding how chemistry-climate model (CCM) biases (which are mainly dynamical) impact simulated ozone columns, and can be used as a benchmark for future UKMO CCM development. The paper is appropriate for publication in GMD, after some revisions (see specific comments below). I also encourage the authors to consider the following suggestions.

**Suggestions:**

Although the paper's structure is clear, I think "Section 3.1 Metrics" would be better placed after the detailed comparisons of dynamical properties and ozone. Moreover, most metrics calculated are not referred to in the following comparisons of dynamical properties and their impact on ozone. My suggestion would be to split the "Results" section into two sections, i.e. "evaluation of stratospheric dynamics and ozone", and "Quantitative assessment, i.e., metrics".

We thank the reviewer for this suggestion. However, it is our feeling that it is better to benchmark the model first, since this allows for more efficient discussion of the in depth diagnostics which follow in Section 3.2. This section actually refers back to the first 6 of the 14 metrics, and is structured to discuss them in order (temperature, wind, and upwelling). Were we to re-order Section 3, then the discussion of dynamics would need to contain more detail on the model biases – detail which would then need to be repeated in the metrics section. Thus, we prefer to keep the structure of the paper as it currently stands.

*No changes required to manuscript.*

**More could be made of the differences in model behaviour between REF-C1 and REF-C2. REF-C1 is usually closer to observations than REF-C2, as expected.**

We agree, and note that this is particularly relevant to Figures 7, 14, 15, and 17. Discussion of the differences between REF-C1 and REF-C2, and the fact that REF-C1 is closer to observations than REF-C2 in these figures, has now been added to the manuscript.

*Changes in manuscript: The following text has been added to the manuscript, in paragraph 1 of Section 3.2.2 (P10L4–7): "In all months, tropical tropopause temperature and water vapour concentrations in REF-C1 are closer to the observations than those in REF-C2 (Figure 7). This may be expected, since REF-C1 is an atmosphere only simulation, and thus forcing from sea surface temperatures will be inline with observations, whereas REF-C2 is a coupled atmosphere-ocean simulation.", in paragraph 1 of Section 3.3.1 (P12L21–22): "Whilst REF-C1 simulates a more accurate phase than REF-C2, errors are most pronounced from $60^\circ E$ to $30^\circ W$, where TCO is too high at $60^\circ S$.", and in paragraph 1 of Section 3.3.2 (P13L17–18): "As noted in Figure 7, the bias in REF-C1 is smaller than that in REF-C2.".*

**Specific comments:**

**1) "Ozone concentrations" appear throughout the paper, but the authors only show total column ozone (TCO). So the authors should replace all "ozone concentrations" with TCO. They are not the same, therefore should not be mixed.**

This is true for all figures except Figure 16. "Ozone concentrations" has been globally replaced with "TCO", as suggested, everywhere except in discussion of Figure 16.

*Changes in manuscript: The text "ozone concentrations" has been replaced with "TCO" everywhere except in discussion of Figure 16.*

**2) In the abstract, the last sentence says that "…that the nudged models still remain far from perfect": Could you elaborate in which sense these models are "far from perfect"? I suggest to re-phrase this statement, and point out any potential problems in applying nudging techniques. It feels like an empty statement to me.**

The abstract has been re-arranged to make clearer the issues that this statement refers to. In addition, "far from perfect" now reads "issues can remain in the climatology of nudged models".

*Changes in manuscript: The end of the abstract (P1L9–12) now reads: "Whilst nudging can, in general, provide a useful tool for removing the influence of dynamical biases from the evolution of chemical fields, this study shows that issues can remain in the climatology of nudged models. Significant biases in stratospheric vertical velocities, age of air, water vapour and total column ozone still exist in the Met Office nudged model. Further, these lead to biases in the downward flux of ozone into the troposphere."*

**3) P3L22: the previous version used in CCMVal2 did have interactive lightning NOx emissions and interactive wet deposition although for a much more limited range of species. Dry deposition used offline tabulated deposition velocities (Morgenstern et al., 2009). Please correct.**

Corrected.

*Changes in manuscript: Relevant text in paragraph 3 of Section 2 (P3L21–27) revised to read: "… interactive lightning emissions are scaled to give 5TgN/yr (O'Connor et al., 2014). … The deposition schemes have been improved since the Met Office's CCMVal-2 configuration, with interactive wet deposition now applied to a wider range of species, and the tabulated dry deposition scheme replaced by a resistance-in-series approach (O'Connor et al., 2014)."*

**4) P6, paragraph 3, you state that nudged simulations do not perform well in metrics of "tropical upwelling and QBO"; could you elaborate on any inconsistencies in treating model's dynamics in nudging and their impact on some simulated model properties? You may want to mention the idea that wind fields used for nudging may not satisfy the continuity equation, which will negatively impact vertical velocity fields.**

Details on how vertical velocity may be negatively impacted by nudging have been added to the text.

*Changes in manuscript: The following text has been added to paragraph 8 of*

*Section 3.1 (P6L26–29): "If the nudged u and v winds do not have zero horizontal divergence then they will force spurious gravity and acoustic modes that will be reflected in spurious vertical velocities. Furthermore, if u and v are not in geostrophic balance then the nudging will introduce ageostrophic motions."*.

**5) Section 3.3.1 "Extratropics" only covers high-latitude aspects. I suggest to either re-title the section to "High latitudes" or give some coverage to mid-latitude aspects.**

The sections on "Extratropics" have been re-titled "High latitudes", as suggested.

*Changes in manuscript: Sections 3.2.1 (P8L19) and 3.3.1 (P12L7) have been re-titled "High latitudes".*

**6) P11L24: Replace "ozone depletion" with total column ozone (TCO, the standard notation). You're not actually quantifying ozone depletion, just total columns. Also L25: Replace "column ozone concentrations" with TCO.**

Changed.

*Changes in manuscript: Text in the first paragraph of Section 3.3.1 (P12L8–10) has been modified to read: "The change in TCO in the extratropics, during the period 1980–2010, is similar in all simulations (Figure 12(c,d)), and agrees well with the TOMS observations. However, TCO that is too high is indicative of an ozone hole that is too small in area.".*

**7) P12L12: That is technically correct, but imposing zonally invariant ozone would not improve the situation. Rather than imposing zonally invariant ozone (which would be inconsistent with best understanding of the ozone distribution), would it be more effective to work on the model to improve the factors that influence the phase of these planetary waves, such as orographic forcing? The discussion of how to impose ozone in models that cannot get the phase of the waves correct strikes me as somewhat missing the point.**

The reviewer is correct that, ideally, the best thing to do is to endeavour to improve the simulated phase of stationary waves in climate models. However, our discussion centers around the pragmatic issue of whether it is best to impose zonally symmetric or zonally asymmetric ozone in current climate models. We have modified the text to make this clearer.

*Changes in manuscript: Last sentence in paragraph 2 of Section 3.3.1 (P12L32–34) changed to read: "In the absence of improvement to the*

*simulated phase of stationary waves, the results here show that prescribing zonally asymmetric ozone will almost always lead to different TCO from those obtained by the same model using self determined ozone.".*

**8) L12L25: I noticed that there is a negative trend in tropical ozone in all simulations, but there does not appear to be much trend in the observations. Please comment on this.**

This is consistent with the findings of the WMO 2010 ozone assessment report. A comment has been added to the text.

*Changes in manuscript: The first paragraph in Section 3.3.2 (P13L13–16) has been modified to read: "The simulated interannual variability in tropical TCO (Figure 12(b)), in both free-running and nudged simulations, agrees well with the observations. However, all simulations show a $\sim 6$ DU reduction in TCO over the period 1980–1995 which is much larger than the observed reduction of $\sim 2$ DU (consistent with Figure 3-6(a) from Chapter 3 of WMO, 2011). Furthermore, TCO is again biased high, …".*

**9) P13L2: It is true that convection, lightning emissions, and BB could impact tropospheric ozone, but they are unlikely the main cause for the 10 DU bias in TCOs here. Actually figure 17b suggest that it's mainly the tropopause height whose variations give you differences in TCO between the simulations. In the troposphere, to partial columns go in parallel (implying there is no significant difference in tropical tropospheric ozone between the simulations). I think your suggestion that tropospheric processes cause this high bias is insufficiently supported by your findings. If this were purely a tropospheric problem, 10 DU would likely amount to an unrealistic 50% error in tropical tropospheric ozone. More likely, it is due mainly to an error in the placement of the tropical tropopause, which you could establish.**

We accept that discussion of convection, lightning emissions, and BB is insufficiently supported, and have removed this discussion from the manuscript. However, Figure 1 in this response to reviewers shows that the model tropopause is at the correct height, and thus is not the cause of errors in TCO.

*Changes in manuscript: The text "where convection, lightning emissions and biomass burning emissions also have an important influence on TCO (Stevenson et al., 2006)" has been removed from the manuscript (P13L23).*

**10) P13L21: Morgenstern et al. (2009) is a more appropriate reference here. This problem was not specifically addressed in Morgenstern et al. (2010).**

Reference changed.

*Changes in manuscript: Reference to Morgenstern et al. (2010), in paragraph 3 of Section 4 (P14L10), changed to Morgenstern et al. (2009).*

**11) P14L7 (cf. Figure 14): That's surprising, considering there should be a close correspondence between the size of the polar vortex, as defined by a transport barrier, and the ozone hole (which is bounded by that transport barrier). If despite nudging these two still differ, could it be that the reanalyses are insufficiently constrained by observations during winter/spring over Antarctica? Please elaborate on the role of the transport barrier in this.**

Here the "ozone hole" is defined as the area over which TCO drops to below 220DU. In HadGEM3-ES, there is a high bias in TCO throughout the tropics and southern high latitudes, and thus the ozone hole will appear too small, regardless of how accurately the barriers to transport are simulated. The definition of ozone hole which is used here has been clarified in the text.

*Changes in manuscript: The text in paragraph 5 of Section 4 (P14L27–32) has been modified to read: "... the high ozone biases that exist in the tropics and southern high latitudes of the free-running model persist also in the nudged model, and these are therefore not solely attributable to biases in the dynamical fields. Thus, despite the fact that the area of southern hemisphere Polar Stratospheric Clouds is correctly simulated in the nudged model, the ozone hole area, defined as the area over which TCO drops to below 220DU, is too small in both free-running and nudged models (an issue which is not unique to HadGEM3-ES, as shown by Figure 1 of Austin et al., 2010).".*

**12) P14L17: Your analysis does not imply errors in any of these processes. To make such a statement, you would have had to compare tropospheric ozone against observations. See above on the role of the tropopause height.**

As above, discussion of convection, lightning emissions, and BB has been removed.

*Changes in manuscript: The text "(and potentially errors in e.g. convection, lightning emissions, and biomass burning emissions and their distribution; Stevenson et al., 2006)" has been removed from the manuscript (P15L2).*

**13) P14L27: You did not directly compare this model version against other models, so I suggest to remove this half-sentence.**

Agreed.

*Changes in manuscript (P15L11): Half-sentence removed.*

[Figure]

Figure 1: Annual climatological mean temperature (10°S–10°N), for the years 1980–2010. The height of the tropopause in both free-running and nudged simulations is 100hPa, consistent with ERA-Interim.